# Handling Learnwares from Heterogeneous Feature Spaces with Explicit Label Exploitation

**Peng Tan, Hai-Tian Liu, Zhi-Hao Tan, Zhi-Hua Zhou**
National Key Laboratory for Novel Software Technology, Nanjing University, China
School of Artificial Intelligence, Nanjing University, China
{tanp,liuht,tanzh,zhouzh}@lamda.nju.edu.cn

## Abstract

The learnware paradigm aims to help users leverage numerous existing high-performing models instead of starting from scratch, where a learnware consists of a well-trained model and the specification describing its capability. Numerous learnwares are accommodated by a learnware dock system. When users solve tasks with the system, models that fully match the task feature space are often rare or even unavailable. However, models with heterogeneous feature space can still be helpful. This paper finds that *label information*, particularly model outputs, is helpful yet previously less exploited in the accommodation of heterogeneous learnwares. We extend the specification to better leverage model pseudo-labels and subsequently enrich the unified embedding space for better specification evolvement. With label information, the learnware identification can also be improved by additionally comparing conditional distributions. Experiments demonstrate that, even without a model explicitly tailored to user tasks, the system can effectively handle tasks by leveraging models from diverse feature spaces.

## 1 Introduction

The current machine learning paradigm has achieved remarkable success across various domains. This success, however, hinges on several critical factors: access to abundant high-quality labeled data, expensive computational resources, and deep expertise in feature engineering and algorithm design. These requirements pose a significant challenge for ordinary individuals aiming to build high-quality models from scratch. Moreover, issues such as data privacy, the difficulty of model adaptation, and catastrophic forgetting complicate the reuse or adaptation of trained models across different users.

Indeed, most efforts have focused on these issues separately, paying less attention to the fact that these problems are entangled. To address these challenges simultaneously, the *learnware* paradigm was proposed by Zhou [2016]. The learnware paradigm [Zhou and Tan, 2024] aims to assist users in solving their tasks by leveraging existing high-performing models, through the establishment of a *learnware dock system*. One important purpose of learnware paradigm is to enable high-performing models, submitted by developers, to be used "beyond-what-was-submitted." This means that models can be repurposed to assist with tasks not originally targeted by developers. To achieve this, learnware is designed as a high-performing model with a *specification* describing its capability and utility. The specification, a central component for learnware management and identification, can be implemented by sketching the data distribution in which the model is proficient [Zhou and Tan, 2024]. Recently, to facilitate research on the learnware paradigm, the learnware dock system, Beimingwu, has been released [Tan et al., 2024a].

Previous research [Liu et al., 2024, Xie et al., 2023, Zhang et al., 2021] focuses on the homogeneous case where models and user tasks share the same feature space. However, in real-world scenarios, the feature spaces of models often differ due to varied feature engineering. As an example, we

consider the widely used clinical database, the OMOP Common Data Model [Biedermann et al., 2021], as illustrated in Figure 1. This model manages healthcare data from various sources through several standardized tables, such as demographic information, diagnoses, laboratory results, and medications. Experts across different hospitals often use different tables for feature engineering, even when working on the same clinical task, leading to the development of heterogeneous models.

In order to manage and exploit models developed from heterogeneous feature spaces, it is essential to build connections between these different spaces. Existing related techniques for exploiting relationships between feature spaces either rely on raw data [Wang and Sun, 2022, Zhu et al., 2023] of the model or utilize additional co-occurrence data [Xu et al., 2013, Huang et al., 2023]. However, with model specifications, the learnware dock system can determine the relationships through subspace learning without the need for raw data or extra auxiliary data [Tan et al., 2023]. To effectively accommodate heterogeneous learnwares, a unified subspace is constructed based on specifications of all submitted models, which helps to evolve the specification to have the capabilities of meeting requirements across different feature spaces. This paper finds that, without label information, subspace learning tends to yield suboptimal results, causing embeddings with entangled

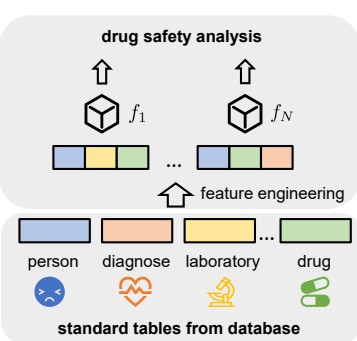

Figure 1: Heterogeneous feature space models in real-world scenario.

class representations in the subspace, or even rendering them meaningless when feature spaces are only weakly correlated. Additionally, without exploiting label information, the system can only identify models with marginal distributions similar to the user's task, ignoring models' capabilities.

This paper explicitly leverages label information for managing and utilizing heterogeneous models. We extend the specification to better incorporate model pseudo-labels, enabling the transition from unsupervised to supervised subspace learning for better specification evolvement. The extended specification also allows for additional comparison of conditional distributions using label information, thereby improving the learnware identification. The contributions are as follows:

- This paper proposes to exploit the model outputs to evolve specifications into a unified space during heterogeneous learnware accommodation. Specifically, the unified space is constructed based on the specifications of all models. By exploiting model outputs encoded in the specification, the resulting subspace exhibits improved properties, with less entangled class representations and more coherent embeddings.

- This paper extends the specification implementation to more effectively leverage label information by encoding both marginal and conditional distributions. This extended specification provides more accurate label information during subspace learning to better evolve specifications. Additionally, it also allows for additional comparison of conditional distributions, thereby improving the learnware identification.

- Experiments demonstrate that, even without a model explicitly tailored to the user's task, the system can effectively handle the task by leveraging models from diverse feature spaces.

## 2 Preliminary

Specification is the central part of the learnware, capturing the model ability. This section briefly introduces the Reduced Kernel Mean Embedding (RKME) specification [Zhou and Tan, 2024], which sketches the joint distribution of task features and model outputs with kernel methods.

We start by introducing the Kernel Mean Embedding (KME) [Schölkopf and Smola, 2002], which offers a novel representation for distributions. KME transforms a distribution into a reproducing kernel Hilbert space (RKHS). Given a distribution $\mathcal{D}$ defined over a space $\mathcal{X}$, the KME is defined as $\mu_k(\mathcal{D}) := \int_{\mathcal{X}} k(\boldsymbol{x}, \cdot) \mathrm{d}\mathcal{D}(\boldsymbol{x})$, where $k : \mathcal{X} \times \mathcal{X} \to \mathbb{R}$ is a symmetric and positive definite kernel function, and its associated RKHS is $\mathcal{H}$. For a data set $\{\boldsymbol{x}_i\}_{i=1}^m$ sampled from $\mathcal{D}$, the empirical estimate of KME is given by $\hat{\mu}_k(\mathcal{D}) := \frac{1}{m} \sum_{i=1}^m k(\boldsymbol{x}_i, \cdot)$.

KME is considered as a potential specification due to several favorable properties. Accessing the raw data, however, compromises the necessary privacy concerns of the specification. Based

on KME, the `RKME` specification is proposed to use a reduced set of minor weighted samples $\{(\beta_j, \boldsymbol{t}_j)\}_{j=1}^n, n \ll m$ to approximate the empirical KME of the original dataset with model pseudo-outputs $\{\boldsymbol{q}_i\}_{i=1}^m = \{(\boldsymbol{x}_i, \hat{y}_i)\}_{i=1}^m$, where $\hat{y}_i = f(\boldsymbol{x}_i)$ is the model prediction. The reduced set is generated by:

$$\min_{\boldsymbol{\beta}, \boldsymbol{t}} \left\| \frac{1}{m} \sum_{i=1}^m k(\boldsymbol{q}_i, \cdot) - \sum_{j=1}^n \beta_j k(\boldsymbol{t}_j, \cdot) \right\|_{\mathcal{H}}^2, \tag{1}$$

with the non-negative coefficients $\{\beta_j\}_{j=1}^n$. The RKME $\Phi(\cdot) = \sum_{j=1}^n \beta_j k(\boldsymbol{t}_j, \cdot) \in \mathcal{H}$ acts as the specification, and the RKHS $\mathcal{H}$ is referred to as the specification space. This specification effectively captures the major information of the distribution $\mathcal{D}$ without exposing raw data and explicitly encodes the model capability based on its outputs. Notably, in simple cases where the features are sufficient to represent the model capability, the sketch solely on the features $\{\boldsymbol{x}_i\}_{i=1}^m$ can also be used as the model specification [Wu et al., 2023]. In this paper, we further extend the specification generation process to more effectively encode the model's outputs.

## 3   Problem setup

This paper addresses the challenge of constructing a heterogeneous learnware dock system and leveraging it to assist users who have only limited labeled data such that training a model by themselves will lead to poor performance. Without loss of generality, we consider the underlying full feature space, denoted as $\mathcal{X}_{\text{all}}$, as a composite of $Q$ distinct blocks, i.e., $\mathcal{X}_{\text{all}} = \mathcal{X}_1 \times \cdots \times \mathcal{X}_Q$. The feature spaces for developers, $\mathcal{X}^{\text{dev}}$, and for users, $\mathcal{X}^{\text{user}}$, are represented as Cartesian products of specific blocks $\times_{i \in C} \mathcal{X}_i$, where $C$ refers to block indices.

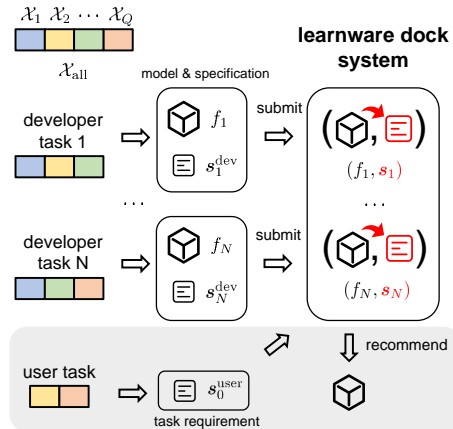

The overall procedure consists of two stages: the submission stage and the deployment stage. In the submission stage, the developer trains a well-performing model $f_i$ on the dataset $D_i := \{(\boldsymbol{x}_{ij}, y_{ij})\}_{j=1}^{n_i}$ and generates a *developer-level* specification $\boldsymbol{s}_i^{\text{dev}}$, which captures the model's performance without exposing raw data. After receiving all heterogeneous models and their developer-

Figure 2: An illustration of the learnware paradigm with heterogeneous feature spaces

level specifications, the learnware dock system assigns a *system-level* specification $\boldsymbol{s}_i$ to each model $f_i$, based on all submitted specifications $\{\boldsymbol{s}_i^{\text{dev}}\}_{i=1}^N$. The heterogeneous learnware dock system is then constructed as $\{(f_i, \boldsymbol{s}_i)\}_{i=1}^N$. In the deployment stage, the user has unlabeled data $D_0^u = \{\boldsymbol{x}_{0i}\}_{i=1}^{n_u}$ and a limited amount of labeled data $D_0^l = \{(\tilde{\boldsymbol{x}}_{0i}, y_{0i})\}_{i=1}^{n_l}$ (the unlabeled data cover labeled data features, i.e., $\{\tilde{\boldsymbol{x}}_i\}_i \subseteq \{\boldsymbol{x}_i\}_i$). The user generates a user-level task requirement $\boldsymbol{s}_0^{\text{user}}$ and submits it to the learnware dock system. The system then identifies the most helpful model(s) for reuse to tackle the user task.

## 4   Methodology

This section outlines our methodology for accommodating heterogeneous models under the learnware paradigm and assisting user tasks, emphasizing the importance and utilization of label information, which remains *unexplored* in learnware paradigm when dealing with heterogeneous feature spaces.

### 4.1   Improve managing heterogeneous models with label information

To handle learnwares with heterogeneous feature spaces, it is helpful to exploit the relationships between these spaces. A common approach is to learn a unified subspace. However, without label information, the resulting subspace may produce entangled embeddings of samples from different

classes, and when feature blocks are weakly correlated, the subspace may become meaningless (see Section B.2 for detailed discussion). Since subspace learning is based on all learnware specifications, incorporating label information into the model specification is highly beneficial.

We rewrite the RKME specification represented by $R = (\boldsymbol{\beta}, T) = \{(\beta_j, \boldsymbol{t}_j)\}_{j=1}^m$ to $\text{RKME}_\text{L}$ represented by $R_L = (\boldsymbol{\beta}, Z, Y) = \{(\beta_j, \boldsymbol{z}_j, y_j)\}_{j=1}^m$ by splitting the sample $\boldsymbol{t}_j$ into the feature $\boldsymbol{z}_j$ and the pseudo label $y_j$, emphasizing label information. Given existing model specifications $\{\boldsymbol{s}_i^{\text{dev}} := \{(\beta_{ij}, \boldsymbol{z}_{ij}, y_{ij})\}_{j=1}^{m_i}\}_{i=1}^N$, the learnware dock system can learn a unified subspace $\mathcal{X}_{\text{sub}}$ with encoding functions $\{h_k : \mathcal{X}_k \mapsto \mathcal{X}_{\text{sub}}\}_{k=1}^Q$ and decoding functions $\{g_k : \mathcal{X}_{\text{sub}} \mapsto \mathcal{X}_k\}_{k=1}^Q$ by optimizing $L = \alpha_1 L_{\text{reconstruction}} + \alpha_2 L_{\text{similar}} + \alpha_3 L_{\text{supervised}}$ over mapping functions $\{h_k, g_k\}_{k=1}^Q$. The objective function has three components: the reconstruction loss, which trains mapping functions $(h_k, g_k)$ to map and reconstruct data in $\mathcal{X}_k$; the similarity loss, which makes embeddings of different slices of $\boldsymbol{z}_j$ similar; and the supervised loss, which uses label information to improve subspace learning by making class embeddings more separable or aligning samples within the same class. After subspace learning, the mapping functions $\{h_k, g_k\}_{k=1}^Q$ can project data from any combination of feature space blocks to the subspace. When reusing heterogeneous models, these functions can also fill in missing parts needed for model predictions. Such a framework can be implemented by existing subspace learning methods, such as self-supervised learning [Ucar et al., 2021, Bahri et al., 2022], matrix factorization [Xu and Gong, 2004, Wang et al., 2016]. When the system receives learnwares from unseen feature spaces after subspace generation, the system can update the subspace during idle time.

## 4.2 Improve matching model and user task with label information

**Matching with only marginal distribution $P_X$ is not enough.** We first review previous methodologies for matching a user's task with a model in the homogeneous case, where all models and user tasks share the same feature space [Wu et al., 2023, Zhang et al., 2021]. These methods recommend the

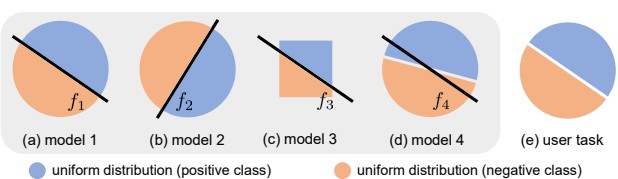

(a) model 1    (b) model 2    (c) model 3    (d) model 4    (e) user task

● uniform distribution (positive class)    ● uniform distribution (negative class)

Figure 3: Label information is beneficial for matching.

model with the most similar marginal distribution $P_X$. To avoid exposing raw data, they use RKME to sketch the marginal distributions of the model task and user task, serving as the model specifications and user requirements. To illustrate the deficiency, we refer to Figure 3, which presents five tasks with uniform distributions. Among these tasks, four have circular support sets and one has a square support set. The two problems are: 1) Models with the same $P_X$ but different $P_{X|Y}$ are indistinguishable. In Figure 3, Models 1, 2 and 4 are all recommended, but model 2 is unsuitable for the user task. 2) Models with different $P_X$ are rarely considered, despite their potential usefulness. Model 3 in Figure 3, though suitable, are excluded because its $P_X$ is square instead of circle.

**Enhance matching by incorporating the conditional distribution $P_{X|Y}$.** Matching solely on the marginal distribution is insufficient for model identification. To better recommend models to user tasks, we propose additionally considering the conditional distribution $P_{X|Y}$, which helps exclude the model with dissimilar conditional distributions (Model 2) and include the model with similar ones (Model 3). While the user's task can estimate the conditional distribution from labeled data, a key question arises for the model task: *should we use true labels or model-generated pseudo labels*? Using True labels results in comparing the user's task distribution $P(X, Y)$ with the model's original task distribution, while pseudo labels results in comparing the model-generated joint distribution $P(X, \hat{Y})$ with new tasks, allowing the model to be reused beyond its original purpose. As shown in Figure 3, Model 4 would be recommended using pseudo labels but not using true labels. In conclusion, considering both marginal and conditional distributions improves model identification, with model-generated pseudo labels being helpful for encoding model capabilities.

## 4.3 Summary

To accommodate and identify models developed from heterogeneous feature spaces, it is advantageous to utilize pseudo-label information generated by the models. To better incorporate this information,

we propose to integrate both marginal and conditional distributions into the model specification and user requirements, represented as $\{(\beta_j, \boldsymbol{z}_j, y_j)\}_{j=1}^m$. By comparing these distributions, we improve learnware identification. The inclusion of label information enhances subspace learning, and the framework can be applied across various subspace learning methods.

# 5   Detailed solution

In this section, we provide the detailed procedure for the heterogeneous learnware problem based on the aforementioned methodology, which consists of model specification generation, heterogeneous learnwares accommodation by the system, and system exploitation for solving new user tasks.

## 5.1   The developer generates the model specification

The model specification sketches task distribution and model capabilities with a reduced set. Instead of sketching the joint distribution of task features and outputs (Eq. (1)), we propose to generate feature and label part separately to balance label and feature information. This includes a unified mechanism for classification and regression, and a specialized mechanism for classification.

**Unified mechanism for classification and regression tasks.**   Given a dataset $D = \{(\boldsymbol{x}_i, y_i)\}_{i=1}^n$ and a model $f$ trained on it, we first generate a reduced set $\{(\beta_j, \boldsymbol{z}_j)\}_{j=1}^m$ solely on $\{\boldsymbol{x}_i\}_{i=1}^n$ based on RKME via Eq. (1) with $\boldsymbol{q}_i = \boldsymbol{x}_i$, which sketches the marginal distribution of the task feature. To encode the model's ability, pseudo labels can be assigned to the reduced set using $y_j = f(\boldsymbol{z}_j)$, resulting in the labeled reduced set $R_L = (\boldsymbol{\beta}, Z, Y) = \{(\beta_j, \boldsymbol{z}_j, y_j)\}_{j=1}^m$, serving as the developer-level model specification $\boldsymbol{s}^{\text{dev}}$.

**Specialized mechanism for classification tasks.**   For classification problems, given the finite label space $\mathcal{Y}$, we propose the mechanism to directly sketch the model's capacity, characterized by the conditional distribution $P(X|Y)$ of the model $f$. We first obtain the model predictions $\{\hat{y}_i\}_{i=1}^n$ on its "skilled" marginal distribution, i.e., its training data $\{\boldsymbol{x}_i\}_{i=1}^n$. Then, the pseudo-labeled dataset $\{(\boldsymbol{x}_i, \hat{y}_i)\}_{i=1}^n$, which encodes the model's conditional distribution, can be sketched by a labeled reduced set $R_L = (\boldsymbol{\beta}, Z, Y) = \{(\beta_j, \boldsymbol{z}_j, y_j)\}_{j=1}^m$ using the following objective:

$$
\left\| \sum_{i=1}^n \frac{1}{n} k\left(\boldsymbol{x}_i, \cdot\right) - \sum_{j=1}^m \beta_j k\left(\boldsymbol{z}_j, \cdot\right) \right\|_{\mathcal{H}_k}^2 + \theta \sum_{c=1}^C \left\| \sum_{i \in \mathcal{I}_c} \frac{1}{n} k\left(\boldsymbol{x}_i, \cdot\right) - \sum_{j \in \mathcal{I}_c'} \beta_j k\left(\boldsymbol{z}_j, \cdot\right) \right\|_{\mathcal{H}_k}^2, \quad (2)
$$

where $\mathcal{I}_c$ and $\mathcal{I}_c'$ represent the indices of samples $\boldsymbol{x}_i$ and $\boldsymbol{z}_j$ belonging to class $c$, respectively. $\theta$ is the parameter used to balance the marginal distribution distance and conditional distribution distance. The labeled reduced set $R_L$ should approximate both the marginal distribution $\sum_{i=1}^n \frac{1}{n} \delta_{\boldsymbol{x}_i}$ with $\sum_{j=1}^m \beta_j \delta_{\boldsymbol{z}_j}$ and the conditional distribution given the $c$ class $\sum_{i \in \mathcal{I}_c} \frac{1}{n} \delta_{\boldsymbol{x}_i}$ with $\sum_{j \in \mathcal{I}_c'} \beta_j \delta_{\boldsymbol{z}_j}$ simultaneously. Here, $\delta(\cdot)$ is the Dirichlet function, which describes the probability mass at a single point. The objective Eq. (2) can be optimized by alternating optimization, the details are showed in E.1. The optimized reduced set $R_L$ is served as the developer-level model specification $\boldsymbol{s}^{\text{dev}}$.

The first unified mechanism sketches the marginal distribution and then encodes the model information, while the second specialized mechanism directly sketches the model's conditional distribution.

## 5.2   The system accommodates heterogeneous learnwares

After the developer-level specification $\boldsymbol{s}^{\text{dev}}$ is generated, the developer submits the model $f$ with specification $\boldsymbol{s}^{\text{dev}}$ to the learnware dock system. The system exploits the relationship of different feature spaces and manages heterogeneous models by assigning system-level specification $\boldsymbol{s}^{\text{sys}}$.

**Subspace learning**   After the learnware dock system receives heterogeneous models with their developer-level specifications, it generates a unified subspace $\mathcal{X}_{\text{sub}}$ to connect different feature blocks $\{\mathcal{X}_i\}_{i=1}^Q$ based on all developer-level specifications $\{\boldsymbol{s}_i^{\text{dev}} := \{(\beta_{ij}, \boldsymbol{z}_{ij}, y_{ij})\}_{j=1}^{m_i}\}_{i=1}^N$. During subspace learning, the learnware dock system learns $2Q$ mapping functions: $\{h_k : \mathcal{X}_k \mapsto \mathcal{X}_{\text{sub}}\}_{k=1}^Q$

and $\{g_k : \mathcal{X}_{\text{sub}} \mapsto \mathcal{X}_k\}_{k=1}^Q$. For a particular sample $\boldsymbol{z}_{ij}$, it can be split into several blocks $\{\boldsymbol{z}_{ij}^{(k)}\}_{k \in C_i}$ according to the feature split $\mathcal{X}_{\text{all}} = \mathcal{X}_1 \times \cdots \times \mathcal{X}_Q$. The encoding function $h_k$ produces the embedding of sample slice $\boldsymbol{z}_{ij}^{(k)}$ as $\boldsymbol{v}_{ij}^{(k)}$, and the decoding function $g_k$ reconstructs it to $\mathcal{X}_k$ as $\tilde{\boldsymbol{z}}_{ij}^{(k)}$.

The loss for subspace learning is implemented as follows: **1)** The reconstruction loss, $\mathcal{L}_{\text{reconstruct}} = \sum_{i=1}^N \sum_{j=1}^{m_i} \beta_{ij} \|\tilde{\boldsymbol{z}}_{ij}^{(k)} - \boldsymbol{z}_{ij}^{(k)}\|_{\text{F}}^2$, penalizes the difference between the original sample $\boldsymbol{z}_{ij}^{(k)}$ and the reconstructed sample $\tilde{\boldsymbol{z}}_{ij}^{(k)}$, weighted by the sample importance $\beta_{ij}$. **2)** The supervised loss involves building a simple classifier on $\{\{(\boldsymbol{v}_{ij}, y_{ij})\}_{j=1}^{m_i}\}_{i=1}^N$, where $\boldsymbol{v}_{ij} = \text{mean}(\{\boldsymbol{v}_{ij}^{(k)}\}_{k \in C_i})$, and calculating the prediction loss to make the embeddings of different classes more separable. **3)** The contrastive loss aims to make the embeddings $\{\boldsymbol{v}_{ij}^{(k)}\}_{k \in C_i}$ of a single sample $\boldsymbol{z}_{ij}$ similar, while ensuring that embeddings of different samples are dissimilar. The contrastive loss $\mathcal{L}_{\text{contrastive}} = \sum_{i=1}^N l_i$ includes $N$ terms, each term being a weighted loss extended from the Self-VPCL loss [Wang and Sun, 2022], calculated on the embeddings $\{\{\boldsymbol{v}_{ij}^{(k)}\}_{k \in C_i}\}_{i=1}^{m_i}$ of one specification $\boldsymbol{s}_i^{\text{dev}}$: $l_i = \sum_{j=1}^{m_i} \beta_{ij} \sum_{k \in C_i} \sum_{k' \in C_i, k' \neq k} \log \frac{\exp \psi\left(\boldsymbol{v}_{ij}^{(k)}, \boldsymbol{v}_{ij}^{(k')}\right)}{\sum_{t=1}^{m_i} \sum_{k^\dagger \in C_i} \exp \psi\left(\boldsymbol{v}_{ij}^{(k)}, \boldsymbol{v}_{it}^{(k^\dagger)}\right)}$, where $\psi$ is the cosine similarity function. In the logarithm term, the numerator represents the similarity of the positive pair, while the denominator is the sum of all pairs. The loss $\mathcal{L}$ is optimized with gradient descent.

**Heterogeneous learnware accommodation.** After subspace learning, the learnware dock system builds a unified subspace and corresponding mapping functions $\{g_i, h_i\}_{i=1}^Q$. The dock system then assigns a system-level specification $\boldsymbol{s}_i = \{(\beta_{ij}, \boldsymbol{v}_{ij}, y_{ij})\}_{j=1}^{m_i}$ for each model based on its developer-level specification $\boldsymbol{s}_i^{\text{dev}} := \{(\beta_{ij}, \boldsymbol{z}_{ij}, y_{ij})\}_{j=1}^{m_i}$. During the system-level specification generation, the sample $\boldsymbol{z}_{ij}$ is projected to the unified subspace as $\boldsymbol{v}_{ij}$, while the coefficient $\beta_{ij}$ and the label $y_{ij}$ remain unchanged. The projection $\boldsymbol{v}_{ij}$ is calculated as follows: $\boldsymbol{v}_{ij} = \frac{1}{|C_i|} \sum_{k \in C_i} h_k(\boldsymbol{z}_{ij}^{(k)})$. The whole procedure of learnware dock system construction is described in Algorithm 3.

## 5.3 The user exploits the learnware dock system

After the learnware dock system accommodates heterogeneous learnwares, users can submit their task requirements to receive recommended models and the toolkit used for feature transformation. They can then directly reuse the model or combine it with a self-training model.

**User requirement generation.** As described in Section 4.2, comparing both the marginal distribution $P_X$ and conditional distribution $P_{X|Y}$ based on the model specification and user requirement helps better learnware identification. Similar to $\text{RKME}_L$ specification encoding both distributions, the $\text{RKME}_L$ requirement of the user is generated similarly to reflect both. In details, given the unlabeled data $D^u = \{\boldsymbol{x}_i\}_{i=1}^{n_u}$ and some labeled data $D^l = \{(\tilde{\boldsymbol{x}}_i, y_i)\}_{i=1}^{n_l}$ (the unlabeled data cover labeled data features, i.e., $\{\tilde{\boldsymbol{x}}_i\}_i \subseteq \{\boldsymbol{x}_i\}_i$), the user can generate $\text{RKME}_L$ requirement presented by labeled reduced set $R_L = \{(\beta_j, \boldsymbol{z}_j, y_j)\}_{j=1}^m$ to sketch the task distribution.

For the classification case, the reduced set $R_L$ can be generated by minimizing the following distance:

$$\left\| \sum_{i=1}^{n_u} \frac{1}{n_u} k(\boldsymbol{x}_i, \cdot) - \sum_{j=1}^m \beta_j k(\boldsymbol{z}_j, \cdot) \right\|_{\mathcal{H}_k}^2 + \theta \sum_{c=1}^C \left\| \sum_{i \in \mathcal{I}_c} \frac{1}{n_l} k(\tilde{\boldsymbol{x}}_i, \cdot) - \sum_{j \in \mathcal{I}_c'} \beta_j k(\boldsymbol{z}_j, \cdot) \right\|_{\mathcal{H}_k}^2 \tag{3}$$

This equation is similar to specification generation in Eq. (2), where the specification sketches the pseudo-labeled dataset, while the requirement sketches semi-supervised data. The first term calculates the distance between the marginal distributions of the unlabeled dataset $\sum_{i=1}^{n_u} \frac{1}{n_u} \delta_{\boldsymbol{x}_i}$ and the reduced set $\sum_{j=1}^m \beta_j \delta_{\boldsymbol{z}_j}$. The second term calculates the distance between the conditional distributions of the labeled dataset $\sum_{i \in \mathcal{I}_c} \frac{1}{n_l} \delta_{\tilde{\boldsymbol{x}}_i}$ and the reduced set $\sum_{j \in \mathcal{I}_c'} \beta_j \delta_{\boldsymbol{z}_j}$, where $\mathcal{I}_c$ and $\mathcal{I}_c'$ denote the sample indices of the labeled dataset and the reduced set with label $c$, respectively. The optimized reduced set $R_L$ becomes the user-level requirement $\boldsymbol{s}_0^{\text{user}}$. The optimization is described in E.2

For regression, the requirement is generated by sketching the marginal distribution and applying a self-trained model for pseudo-labeling, similar to unified specification generation in Section 5.1.

**Learnware identification.** After the user submits the user-level task requirement $s_0^{\text{user}} = \{(\beta_{0k}, z_{0k}, y_{0k})\}_{k=1}^{m_0}$ to the dock system, the dock system transforms it into the system-level task requirement $s_0 = \{(\beta_{0k}, v_{0k}, y_{0k})\}_{k=1}^{m_0}$ by projecting $z_{0k}$ into the subspace as $v_{0k}$. The dock system then calculates the distance between the system-level specification $s_i = \{(\beta_{ij}, v_{ij}, y_{ij})\}_{j=1}^{m_i}$ and the system-level task requirement $s_0$ as follows:

$$\left\| \sum_k^{m_0} \beta_{0k} k(v_{0k}, \cdot) - \sum_j^{m_i} \beta_{ij} k(v_{ij}, \cdot) \right\| + \alpha \sum_C \left\| \sum_{k \in \mathcal{I}_{0,C}} \beta_{0k} k(v_{0k}, \cdot) - \sum_{j \in \mathcal{I}_{i,C}} \beta_{ij} k(v_{ij}, \cdot) \right\|, \quad (4)$$

which measures both the conditional distribution distances and marginal distribution distances. Where $\mathcal{I}_{i,C}$ represents the indices of $v_{ij}$ with the class c. The learnware dock system then recommends the learnware with a minimal distance.

**Learnware reuse.** Once the user receives the recommended model $f_i$ and the dock system toolkit $\{h_k, g_k\}_{k=1}^Q$, they can apply the model to their task. The toolkit helps bridge the gap between different feature spaces. For example, if the user's task is on $\mathcal{X}_1 \times \mathcal{X}_2$ and the model is on $\mathcal{X}_2 \times \mathcal{X}_3 \times \mathcal{X}_4$, the user can project their data using $h_1$ and $h_2$, then decode it to $\mathcal{X}_3$ and $\mathcal{X}_4$ with $g_3$ and $g_4$. The user can use the recommended model's predictions directly or ensemble them with a self-trained model.

### 5.4 Overall procedure

In the submission stage, the dock system receives models with developer-level specifications that sketch model capabilities and assigns system-level specifications by a learned unified subspace. In the deployment stage, users submit task requirements detailing marginal and conditional distributions to receive recommended learnware. This learnware can be integrated with their self-trained models to significantly enhance performance. The overall process is summarized in Appendix D.1.

## 6 Experiments

### 6.1 Experiment setup

**Datasets.** We tested our methods on 30 datasets from the Tabzilla benchmark [McElfresh et al., 2023], excluding tiny datasets. These include 23 classification tasks and 7 regression tasks. For classification tasks, the sample sizes range from 1,000 to 58,310, feature space dimensions from 7 to 7,200, and the number of classes from 2 to 10. For regression tasks, the sample sizes range from 418 to 108,000, and feature space dimensions from 8 to 128.

**Compared methods.** As the heterogeneous learnware problem, where the user has some labeled data, is a new problem, we first compare our approach with two basic methods that train models from scratch: `lightgbm` [Ke et al., 2017], a widely used tree-based method for tabular datasets, and `TabPFN` [Hollmann et al., 2023], a recently proposed prior-data fitted network capable of training and inference on small classification datasets in less than one second. When seeking assistance from the model repository, one simple but inefficient approach is to fetch all models, conduct heterogeneous transfer learning, and select the best one. `Align_unlabel` [Tan et al., 2024a] aligns the feature space and uses the aligned model directly, while `Align_label` [Tan et al., 2024a] goes a step further by fine-tuning through training a new model with augmented features that include aligned model predictions. Another method for reusing knowledge from heterogeneous tasks involves pre-training a unified tabular network on different tables and fine-tuning on the downstream user tasks: `Transtab` [Wang and Sun, 2022] and `Xtab` [Zhu et al., 2023]. However, these methods require access to raw task data, whereas our method protects user privacy. Next, we compare with `Hetero` [Tan et al., 2023], an initial attempt to address the heterogeneous learnware problem without using label information. Finally, we substitute the specification in our method with the RKME specification from [Zhou and Tan, 2024] as `Our_basic` and conduct a comparison with the proposed method.

**Experiment configuration.** The feature space is randomly divided into four equal blocks, creating feature spaces from three-block combinations for developer tasks and two-block combinations for user tasks. For user tasks, 100 labeled data points are sampled from the training set. All experiments are repeated five times. For more details, please see Appendix F.1.

Table 1: Accuracy (%) (mean ± std) on user data true labels. The best performance is in bold.

| Dataset name | Lightgbm | TabPFN | Align$_{unlabel}$ | Align$_{label}$ | Transtab | Xtab | Hetero | Our$_{basic}$ | Our$_{unify}$ | Our$_{cls}$ |
|---|---|---|---|---|---|---|---|---|---|---|
| credit-g | 67.4 ± 2.1 | 70.5 ± 0.4 | 58.3 ± 2.4 | 69.1 ± 1.6 | 69.6 ± 1.0 | 70.3 ± 0.6 | 67.5 ± 0.0 | 71.2 ± 0.7 | 71.0 ± 1.0 | **71.4 ± 0.7** |
| semeion | 54.8 ± 4.9 | 63.4 ± 1.6 | 6.8 ± 1.9 | 59.6 ± 5.7 | 52.1 ± 8.4 | 27.0 ± 3.4 | 40.2 ± 0.0 | 53.2 ± 0.8 | 59.9 ± 1.6 | **63.7 ± 0.9** |
| mfeat-karhunen | 66.1 ± 14 | 72.6 ± 1.8 | 7.1 ± 3.2 | 73.3 ± 12 | 67.2 ± 1.0 | 36.0 ± 1.9 | 59.4 ± 0.0 | 71.7 ± 2.8 | 73.5 ± 0.2 | **77.6 ± 1.3** |
| splice | 64.9 ± 11 | 63.2 ± 0.8 | 42.5 ± 3.1 | 56.2 ± 5.6 | 72.8 ± 1.6 | 56.0 ± 1.2 | 68.4 ± 0.0 | 72.2 ± 2.0 | 73.5 ± 0.9 | **73.6 ± 0.9** |
| gina agnostic | 75.5 ± 1.5 | 76.6 ± 1.7 | 45.1 ± 1.7 | 74.5 ± 1.0 | 78.4 ± 5.2 | 67.8 ± 2.3 | 82.0 ± 0.0 | 88.9 ± 0.5 | 87.3 ± 2.5 | **89.9 ± 0.3** |
| Bioresponse | 64.2 ± 5.7 | 57.4 ± 2.4 | 48.4 ± 2.3 | 64.0 ± 1.6 | 51.6 ± 3.8 | 56.4 ± 2.0 | 66.7 ± 0.0 | 67.1 ± 1.8 | 67.9 ± 2.6 | **71.5 ± 0.9** |
| sylvine | 68.6 ± 20 | 71.4 ± 0.5 | 47.2 ± 4.3 | 69.2 ± 16 | 69.6 ± 0.9 | 66.6 ± 2.2 | 75.1 ± 0.0 | 74.8 ± 1.7 | 74.5 ± 1.8 | **76.9 ± 0.1** |
| christine | 61.7 ± 0.4 | 65.4 ± 1.3 | 46.6 ± 4.7 | 66.9 ± 0.8 | 54.6 ± 4.9 | 61.8 ± 1.2 | 72.2 ± 0.0 | 72.0 ± 0.2 | 72.6 ± 0.2 | **72.7 ± 0.0** |
| theorem-proving | 42.1 ± 1.8 | 42.5 ± 1.0 | 27.0 ± 1.6 | 39.5 ± 1.2 | 42.0 ± 0.6 | 40.8 ± 0.2 | 44.3 ± 0.0 | **51.2 ± 0.0** | 50.4 ± 0.7 | 51.0 ± 0.6 |
| satimage | 80.5 ± 1.5 | 83.3 ± 0.6 | 6.3 ± 4.4 | 81.4 ± 0.9 | 83.5 ± 1.8 | 75.4 ± 2.3 | 48.8 ± 0.0 | 82.8 ± 1.3 | 86.9 ± 0.0 | **87.1 ± 0.2** |
| fabert | 23.4 ± 0.0 | 24.0 ± 0.4 | 14.0 ± 3.5 | 32.0 ± 5.6 | 28.4 ± 5.6 | 22.9 ± 0.5 | 34.2 ± 0.0 | 41.2 ± 1.6 | 41.0 ± 0.0 | **46.0 ± 2.1** |
| gesture segmentation | 40.7 ± 1.4 | 43.1 ± 2.3 | 14.2 ± 1.6 | 37.7 ± 3.4 | 39.7 ± 5.4 | 38.6 ± 2.7 | 37.6 ± 0.0 | **53.6 ± 0.0** | 52.8 ± 0.7 | 52.1 ± 0.8 |
| robert | 22.5 ± 1.3 | 24.3 ± 1.0 | 9.4 ± 0.7 | 24.6 ± 0.8 | 12.0 ± 2.0 | 13.4 ± 1.3 | 45.5 ± 0.0 | 44.8 ± 0.0 | 45.3 ± 0.2 | **45.7 ± 0.2** |
| artificial-characters | 25.0 ± 2.9 | 31.6 ± 0.8 | 8.5 ± 1.7 | 20.2 ± 2.7 | 21.2 ± 1.8 | 21.1 ± 1.7 | 41.4 ± 0.0 | 40.1 ± 1.6 | 41.4 ± 1.2 | **41.8 ± 0.0** |
| eye_movements | 42.8 ± 2.9 | 42.1 ± 0.5 | 31.9 ± 1.2 | 39.7 ± 1.6 | 41.0 ± 3.9 | 39.2 ± 1.5 | 46.9 ± 0.0 | 54.9 ± 0.9 | 54.2 ± 0.7 | **55.8 ± 0.0** |
| nursery | 59.1 ± 19 | 60.3 ± 0.9 | 20.4 ± 7.6 | 57.9 ± 17 | 60.2 ± 1.3 | 58.4 ± 1.0 | 57.9 ± 0.0 | **62.2 ± 0.0** | 60.5 ± 1.2 | 62.0 ± 0.0 |
| eeg-eye-state | 59.2 ± 2.2 | 62.8 ± 2.0 | 47.2 ± 2.5 | 55.3 ± 0.3 | 53.7 ± 2.4 | 58.4 ± 1.0 | 63.2 ± 0.0 | 64.4 ± 0.0 | 63.1 ± 1.8 | **65.9 ± 0.0** |
| magic | 73.6 ± 4.7 | 76.2 ± 1.0 | 48.3 ± 4.9 | 72.1 ± 2.5 | 73.6 ± 3.5 | 72.4 ± 2.0 | 62.5 ± 0.0 | 70.6 ± 1.3 | 75.9 ± 0.1 | **76.8 ± 0.2** |
| riccardo | 73.1 ± 1.5 | 75.2 ± 0.4 | 74.9 ± 0.1 | 72.3 ± 2.0 | 74.9 ± 0.2 | 75.0 ± 0.0 | **83.2 ± 0.0** | 81.8 ± 0.5 | 83.1 ± 0.1 | 81.9 ± 0.6 |
| guillermo | 57.7 ± 4.4 | 59.9 ± 0.1 | 58.6 ± 0.6 | 60.0 ± 0.0 | 58.9 ± 0.7 | 59.9 ± 0.1 | 63.6 ± 0.0 | 63.0 ± 0.0 | 62.5 ± 4.6 | **68.4 ± 1.9** |
| nomao | 85.8 ± 2.9 | 86.3 ± 0.9 | 57.5 ± 10 | 86.7 ± 3.4 | 89.0 ± 0.4 | 82.2 ± 1.0 | 79.9 ± 0.0 | 89.6 ± 0.1 | **90.0 ± 0.1** | 89.5 ± 0.2 |
| click prediction | 80.2 ± 1.4 | **83.2 ± 0.0** | 80.1 ± 2.6 | 83.0 ± 0.4 | **83.2 ± 0.0** | 82.6 ± 0.4 | 82.4 ± 0.0 | 80.8 ± 2.0 | **83.2 ± 0.0** | **83.2 ± 0.0** |
| volkert | 39.3 ± 1.7 | 43.0 ± 0.8 | 11.4 ± 2.2 | 42.1 ± 2.1 | 38.1 ± 2.3 | 37.4 ± 1.0 | 40.9 ± 0.0 | 47.9 ± 0.8 | **49.0 ± 1.1** | 46.0 ± 1.5 |
| Avg | 0.577 | 0.599 | 0.353 | 0.581 | 0.572 | 0.530 | 0.593 | 0.652 | 0.661 | **0.674** |
| Our$_{unify}$ w/t/l | 23/0/0 | 20/1/2 | 23/0/0 | 23/0/0 | 22/1/0 | 23/0/0 | 17/1/5 | 13/0/10 | - | 4/1/18 |
| Our$_{cls}$ w/t/l | 23/0/0 | 22/1/0 | 23/0/0 | 23/0/0 | 23/0/0 | 22/1/0 | 22/0/1 | 18/0/5 | 18/1/4 | - |
| Avg. rank | 6.826 | 4.826 | 9.826 | 6.565 | 6.174 | 7.696 | 5.261 | 3.391 | 2.522 | **1.435** |

Table 2: RMSE (mean ± std) on user data true labels. The best performance is emphasized in bold.

| Dataset name | Lightgbm | Align$_{unlabel}$ | Align$_{label}$ | Transtab | Xtab | Hetero | Our$_{basic}$ | Our$_{unify}$ |
|---|---|---|---|---|---|---|---|---|
| pbc | 0.411 ± 0.041 | 0.507 ± 0.011 | 0.402 ± 0.039 | 0.491 ± 0.003 | 0.434 ± 0.010 | 0.396 ± 0.000 | 0.396 ± 0.003 | **0.395 ± 0.005** |
| colleges | 0.194 ± 0.006 | 0.234 ± 0.005 | 0.194 ± 0.007 | 0.249 ± 0.005 | 0.199 ± 0.005 | 0.232 ± 0.000 | 0.196 ± 0.004 | **0.188 ± 0.001** |
| cpu-small | 0.172 ± 0.021 | 0.229 ± 0.006 | 0.168 ± 0.022 | 0.238 ± 0.008 | 2.614 ± 4.923 | 0.211 ± 0.000 | **0.132 ± 0.003** | 0.152 ± 0.035 |
| kin8nm | 0.178 ± 0.014 | 0.204 ± 0.007 | **0.170 ± 0.014** | 0.242 ± 0.006 | 0.179 ± 0.002 | **0.170 ± 0.000** | 0.172 ± 0.001 | **0.170 ± 0.001** |
| dataset-sales | 0.072 ± 0.006 | 0.086 ± 0.005 | 0.068 ± 0.011 | 0.464 ± 0.004 | 0.079 ± 0.003 | 0.103 ± 0.000 | 0.061 ± 0.003 | **0.058 ± 0.001** |
| california | 0.172 ± 0.021 | 0.251 ± 0.005 | **0.163 ± 0.020** | 0.217 ± 0.008 | 0.169 ± 0.004 | 0.197 ± 0.000 | 0.190 ± 0.023 | **0.163 ± 0.022** |
| aloi | 0.303 ± 0.003 | 0.327 ± 0.005 | 0.301 ± 0.006 | 0.322 ± 0.004 | 0.301 ± 0.005 | 0.249 ± 0.000 | **0.224 ± 0.011** | 0.233 ± 0.008 |
| Avg | 0.215 | 0.263 | 0.209 | 0.318 | 0.568 | 0.223 | 0.196 | **0.194** |
| Our$_{unify}$ w/t/l | 7/0/0 | 7/0/0 | 5/2/0 | 7/0/0 | 7/0/0 | 6/1/0 | 5/0/2 | - |
| Avg. rank | 4.286 | 7.143 | 2.571 | 7.429 | 5.286 | 4.286 | 2.714 | **1.286** |

## 6.2 Performance on user tasks

Tables 1 and 2 compare the performance of our proposed methods with other contenders on classification and regression tasks. `Our_unify` refers to the performance of the overall procedure with the unified specification, while `Our_cls` refers to the specialized specification designed for classification tasks. Our approach, `Our_unify`, outperforms the competitors in most cases. While `Lightgbm` and `TabPFN` use self-training, their performance is limited by the small amount of labeled data. It is showed that `TabPFN` performs better than `Lightgbm` under these conditions. This highlights the importance of leveraging well-trained models, even with heterogeneous feature spaces, to improve performance.

Examining `Align_unlabel` shows that heterogeneous transfer learning with only aligning feature spaces without labels is less effective than self-training. However, further fine-tuning enables `Align_label` to outperform self-training methods. Nevertheless, without leveraging knowledge across different tasks, `Align_label` still performs worse than our approach. `Transtab` and `Xtab` attempt to create a unified backbone across different tables to leverage cross-task knowledge, but they fail to reuse the high-performing model on each developer task, leading to worse performance than ours. These methods also require training on raw developer data, whereas our method only accesses model specifications without exposing raw data.

`Hetero` performs worse than our methods due to its lack of modeling the conditional distribution of submitted models and its reliance on unsupervised subspace learning. Compared to `Our_basic`, our proposed specification outperforms the RKME specification, as it alleviates the issue of label information being overshadowed by feature information during specification generation and comparison. Notably, for classification tasks, our specialized model `Our_cls` outperforms `Our_unify` due to its ability to encode conditional distribution of the model more effectively.

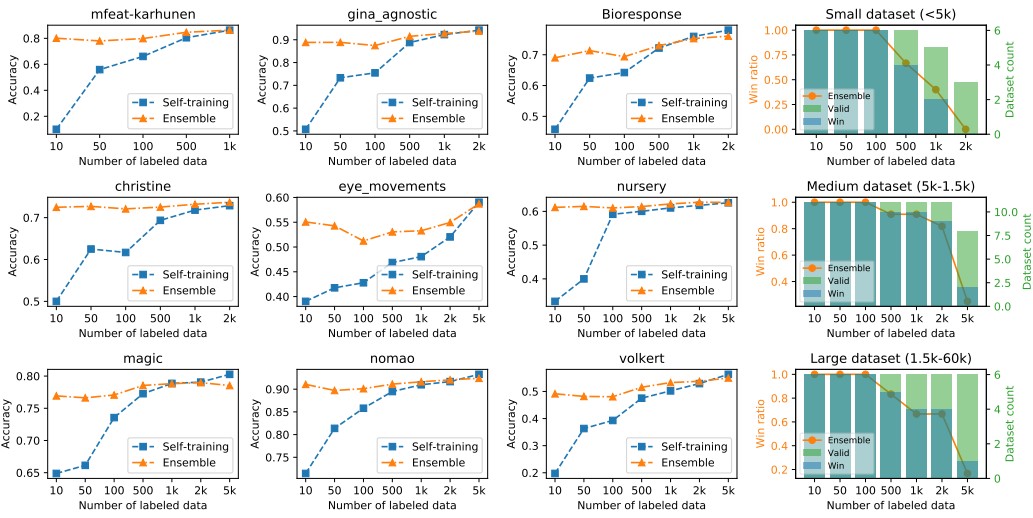

Figure 4: User performance curve for classification tasks.

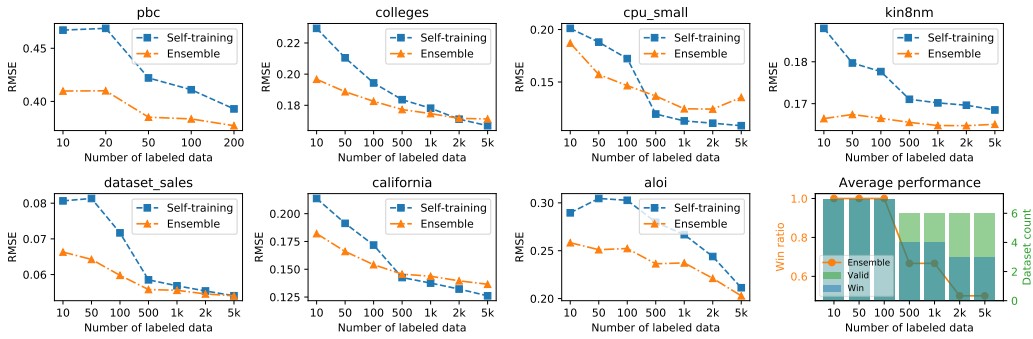

Figure 5: User performance curve for regression tasks.

## 6.3 Evaluation on users with different size of labeled data

In the previous section, we showed that using a single learnware with heterogeneous feature spaces outperforms training models from scratch when labeled data is limited. Now, we analyze how performance changes as users train models and ensemble their predictions with learnware across different amounts of labeled data. Figures 4 and 5 display these trends for classification and regression tasks. These figures indicate that ensemble methods consistently outperform self-training with 100 labeled data points. With 500 labeled data points, the ensemble method still performs better in nearly 80% of cases. Additionally, learnware continues to enhance performance even with 5000 labeled samples, improving 21% of classification cases and 50% of regression cases. For certain datasets like kin8nm in regression tasks, even when users use their entire training dataset, the recommended heterogeneous learnware can still significantly boost performance.

## 7 Conclusion

This paper evolves specifications to a unified subspace with explicit exploitation of model outputs under the heterogeneous learnware scenario. The specification is extended by additionally encoding conditional distribution to better encode the model capability, which can be further evolved by more effective subspace learning enriched by label information. The extended specification also improves learnware identification by additionally matching conditional distributions. We present the complete workflow of the learnware dock system accommodating heterogeneous learnwares and validate the effectiveness of the proposed methods through extensive experiments.

## Acknowledgments

This research was supported by NSFC (62250069) and the Program for Outstanding PhD Candidates of Nanjing University (202401B07). The authors would like to thank Jian-Dong Liu and Jia-Wei Shan for helpful discussions. We are also grateful for the anonymous reviewers for their valuable comments.

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

## Table of contents

## A  Notations

The major notations of this paper are summarized in Table 3.

| Category | Notations | Description |
|---|---|---|
| **basic** | $\mathcal{X}_{\text{all}} = \mathcal{X}_1 \times \cdots \times \mathcal{X}_Q$ | the overall feature space and its $Q$ blocks, the corresponding dimensions are $d, d_1, \cdots, d_Q$. |
| | $\mathcal{X}_1^{\text{dev}}, \cdots, \mathcal{X}_T^{\text{dev}}$ | $T$ kinds of feature spaces for developers' task, each of them is the Cartesian product of several feature blocks. The index set of blocks that $\mathcal{X}_k^{\text{dev}}$ has is $C_k$, i.e., $\mathcal{X}_k^{\text{dev}} = \times_{i \in C_k} \mathcal{X}_i$. |
| | $\mathcal{X}^{\text{user}}$ | the feature space of the user's task, it is the Cartesian product of several feature blocks, the corresponding index set of blocks is $C_0$, i.e., $\mathcal{X}^{\text{user}} = \times_{i \in C_0} \mathcal{X}_i$. |
| | $\mathcal{Y}$ | the label space. |
| **developer** | $D_i := \{(\boldsymbol{x}_{ij}, y_{ij})\}_{j=1}^{n_i}$ | the labeled dataset of the $i$-th developer defined on $\mathcal{X}_{\phi_i}^{\text{dev}} \times \mathcal{Y}$ where $\phi_i \in [1, \cdots, T]$ represents the developer feature space index. |
| | $f_i : \mathcal{X}_{\phi_i}^{\text{dev}} \mapsto \mathcal{Y}$ | the model of the $i$-th developer trained on $D_i$. |
| | $\boldsymbol{s}_i^{\text{dev}} := \{(\beta_{ij}, \boldsymbol{z}_{ij})\}_{j=1}^{m_i}$ | the *developer-level* specification of the $i$-th model generated from $D_i$ via $\mathtt{RKME_L}$. |
| **user** | $D_0^u = \{\boldsymbol{x}_{0i}\}_{i=1}^{n_u}$ | the unlabeled dataset of the user. |
| | $D_0^l = \{(\tilde{\boldsymbol{x}}_{0i}, y_{0i})\}_{i=1}^{n_l}$ | the user's labeled dataset, which is of *limited size* |
| | $\boldsymbol{s}_0^{\text{user}} := \{(\beta_{0j}, \boldsymbol{z}_{0j})\}_{j=1}^{m_0}$ | the *user-level* requirement of the user generated from $D_0$ via $\mathtt{RKME_L}$. |
| **learnware dock system** | $\boldsymbol{s}_i := \{(\beta_{ij}, \boldsymbol{v}_{ij})\}_{j=1}^{m_i}$ | the *system-level* specification of the $i$-th model assigned by the learnware dock system, which is generated by adjusting the developer-level specification $\boldsymbol{s}_i^{\text{dev}}$. |
| | $\boldsymbol{l}_i := (f_i, \boldsymbol{s}_i)$ | the $i$-th learnware accommodated by the learnware dock system. |
| | $\{\boldsymbol{l}_i\}_{i=1}^N$ | the heterogeneous learnware dock system. |
| | $\boldsymbol{s}_0 := \{(\beta_{0j}, \boldsymbol{v}_{0j})\}_{j=1}^{m_0}$ | the *system-level* requirement of the user task generated by the learnware dock system. |
| **subspace** | $\mathcal{X}_{\text{sub}}$ | the learned subspace with dimension $d_{\text{sub}}$. |
| | $h_k : \mathcal{X}_k \mapsto \mathcal{X}_{\text{sub}}$ | the mapping function which projects the data on the $k$-th feature block $\mathcal{X}_k$ to the subspace $\mathcal{X}_{\text{sub}}$. |
| | $g_k : \mathcal{X}_{\text{sub}} \mapsto \mathcal{X}_k$ | the mapping function which reconstructs the data on the subspace $\mathcal{X}_{\text{sub}}$ to the $k$-th feature block $\mathcal{X}_k$. |

Table 3: Major notations of this work.

## B  More discussion

### B.1  Superiority of the learnware paradigm for handling heterogeneous models

**Difficulties of managing models with heterogeneous feature spaces.** To manage models developed from heterogeneous feature spaces, it is essential to exploit the relationship among the corresponding feature spaces $\{\mathcal{X}_i^{\text{dev}}\}_{i=1}^N$. While multi-view learning [Xu et al., 2013] can be beneficial if co-occurrence data across the entire feature space $\mathcal{X}_{\text{all}}$ is available, obtaining such data is nearly impossible in real-world scenarios. Alternatively, if the raw data of the model task is accessible,

training a unified tabular network on heterogeneous tables [Wang and Sun, 2022, Zhu et al., 2023, Yang et al., 2023] offers another approach to reusing knowledge from heterogeneous tasks. However, a unified model often struggles to perform well across all source tasks due to complex and sometimes conflicting internal patterns. Additionally, in sensitive areas like medicine, data sharing is restricted, and privacy concerns prevent access to raw data. *In our problem, raw data is inaccessible to protect the model provider's privacy, and we do not use hard-to-collect auxiliary data.*

**Exploit feature space relationship with model specifications.** Under the learnware paradigm, each model is submitted with a specification that describes its abilities. This specification can be naturally used to explore the relationships between feature spaces. [Tan et al., 2023] generates a unified subspace $\mathcal{X}_{\text{sub}}$ and linear projection functions linking it to all feature blocks $\{\mathcal{X}_i\}_{i=1}^Q$ by leveraging model RKME specifications generated solely on features. However, the specification lack of label information often leads to unsatisfactory performance of subspace learning, like entangled classes of embeddings, it also performs poorly when feature blocks are weakly dependent. This paper better incorporates label information into the specification to improve subspace learning for better heterogeneous learnware accommodation.

### B.2 Exploit label information to handle models with heterogeneous feature spaces

To handle learnwares with heterogeneous feature spaces, it's crucial to exploit the relationship between different feature spaces. When the overall feature space is divided into disjoint blocks and each task data is an arbitrary Cartesian product of several blocks, this exploitation can be divided into two parts: learning relationships between co-occurring feature spaces using data of specific specification, and learning relationships between non-co-occurring feature spaces across all specifications. The first part involves subspace learning to identify a unified subspace, while the second ensures that embeddings of heterogeneous data with intersecting features are closely aligned in the subspace. The first part lays foundation for the second, which is key to managing models with heterogeneous feature spaces.

For the foundational step of subspace learning, if label information is not available, the embeddings of different slices of the same data may not align correctly within the subspace. This misalignment can result in embeddings from different slices having entangled or mixed classes. Furthermore, in extreme cases where feature blocks are jointly independent, subspace learning becomes meaningless due to the irrelevance of features. However, when label information is available, the feature blocks are no longer independent, as all information is used to generate the labels. More discussion that the label information is useful for building connections between independent feature blocks can be found in [Guo et al., 2024].

In summary, label information is essential for subspace learning, as it mitigate entangled classes of embeddings, which critically affects the learnware identification and reuse. It also help ensure performance even when feature blocks are weakly dependent.

## C   Related work

**The learnware paradigm.** The learnware paradigm [Zhou, 2016, Zhou and Tan, 2024] offers a systematic approach to managing well-trained models and leveraging their capabilities to assist users in solving their tasks, rather than training a model from scratch. A learnware consists of a well-trained model accompanied by a specification that describes its capabilities, with this specification being the central component of the learnware. Wu et al. [2023] proposed the RKME specification, which uses a reduced set to sketch the distribution of the task data. Based on the RKME specification, Wu et al. [2023] proposed to match the data distribution for learnware identification, while Zhang et al. [2021] extended it to handle user tasks with unseen parts. To efficiently recommend learnwares among numerous learnwares, Liu et al. [2024] suggested evolving the specification with other learnwares for more accurate identifications and construct the specification index for managing learnwares for efficient learnware search, Xie et al. [2023] proposed using minor representative learnwares as anchors to speed up learnware identification without traversing the whole system.

Previous research has primarily focused on the homogeneous case, where all models and user tasks share the same feature space. However, in real-world applications, the feature spaces of developer models and user tasks often differ. Tan et al. [2024b] was the first to consider the heterogeneous feature space scenario, but it assumes that the original training data is accessible, and auxiliary data

across the entire feature space is collected. To relax this strong assumption of data accessibility, Tan et al. [2023] investigated the organization and utilization of a heterogeneous learnware dock system without requiring access to the original data or auxiliary data across the feature space. While this approach is more realistic, its lack of effective use of label information leads to unsatisfactory performance. This paper examines the importance of label information and integrates it throughout the entire process of the heterogeneous learnware dock system. As a broader impact, the detailed implementation of incorporating label information into the learnware specification can help enhance various aspects of the learnware paradigm. In addition to research on heterogeneous feature spaces, Guo et al. [2023] considered scenarios involving heterogeneous labels.

Based on above research, the first learnware docking system, Beimingwu [Tan et al., 2024a], was recently released. The system streamlines the entire learnware process and provides a highly scalable architecture, facilitating future algorithm implementation and experimental research.

**Related techniques.** To measure the distance between two *labeled* datasets in the same feature space, Alvarez-Melis and Fusi [2020] proposed the Optimal Transport Dataset Distance (OTDD). This approach separately calculates feature and label distances using optimal transport and then combines them. The label distance is derived from the feature distances of partial samples with specific labels. This method aligns with the proposed loss for sketching the labeled dataset to generate specifications and requirements, where the loss consists of feature and label components based on MMD, with the label loss defined by the conditional distributions $P(X|Y)$. Comparing simply concatenating feature and label, separately tackling feature and label can better measure the distance without label information overwhelmed by the longer feature information. For measuring the distance between two distributions in *different feature spaces*, Mémoli [2011] proposed the Gromov–Wasserstein distance, which aggregates all distances of tetrads to measure the distances between two points. In our work, we introduce a method to measure the distance between two labeled datasets in heterogeneous feature spaces using subspace learning and maximum mean discrepancy (MMD) over marginal and conditional distributions.

Existing studies on heterogeneous feature spaces, such as heterogeneous domain adaptation [Duan et al., 2012, Wang and Mahadevan, 2011], heterogeneous transfer learning [Day and Khoshgoftaar, 2017], and heterogeneous model reuse [Ye et al., 2018, 2020], generally map different feature spaces to an intermediate subspace. This process typically requires original data from both domains or co-occurrence data to establish the relationship between different spaces. However, in the learnware paradigm, managing models developed from different feature spaces without auxiliary data becomes feasible due to the existence of RKME specifications associated with each model. Based on this paradigm, we can accommodate, identify, and reuse heterogeneous models of any type without accessing original data or additional co-occurrence auxiliary data. Recently, Guo et al. [2024] explored the relationship between two intersecting feature spaces from a causal perspective, showing that residuals from model predictions can provide information into unobserved variables, specifically, the partial derivative of the true generating function with respect to these unobserved variables. This finding aligns with our approach, where we leverage label information to explore the relationship of different feature spaces.

Recently, some works have focused on identifying and reusing models from a model hub. Ding and Zhou [2020] select models based on an anomaly detector associated with each model, which helps determine whether a feature sample is appropriate for prediction with that model. Ding et al. [2022] propose selecting models using a task-model metric that requires only minimal interaction with data providers. Zhang et al. [2023] propose selecting pre-trained models by calculating the similarity between learned model embeddings and task embeddings, both of which are obtained through the ranking loss. Yi et al. [2024] explore the model selection specifically for visual language models. However, it is important to note that none of these approaches can directly address scenarios involving heterogeneous feature spaces.

# D   Algorithm details

## D.1   Summary of overall procedure

The overall procedure of the heterogeneous learnware dock system consists of two stages. In the submission stage, the dock system receives models with developer-level specifications sketching

model capabilities and assigns system-level specifications using a learned unified subspace. In the deployment stage, users submit task requirements detailing marginal and conditional distributions to receive the recommended learnware. This learnware can be integrated with their self-trained models to significantly enhance performance. The detailed procedures for each stage are outlined in Algorithm 1 and 2, respectively.

---

**Algorithm 1** Submitting stage (learnware accommodation by the system)

1: Each developer trains a model $f_i$ and generates the developer-level specification $s_i^{\text{dev}}$ on the dataset $D_i$ defined on $\mathcal{X}_i^{\text{dev}} \times \mathcal{Y}$.
2: Each developer submits both the model and the developer-level specification $(f_i, s_i^{\text{dev}})$ to the learnware dock system.
3: The learnware dock system generates the projection function $h_i : \mathcal{X}_i \mapsto \mathcal{X}_{\text{sub}}$, reconstruction function $g_i : \mathcal{X}_{\text{sub}} \mapsto \mathcal{X}_i$ *for each feature block* $\mathcal{X}_i$ and the system-level specification $s_i$ for each model $f_i$ based on all developer-level specifications $\{s_i^{\text{dev}}\}_{i=1}^N$.
4: The model $f_i$ is accommodated by the learnware dock system with system-level specifications as learnware $l_i := \{f_i, s_i\}$.
5: The heterogeneous learnware dock system is established as $\{l_i\}_{i=1}^N$.

---

**Algorithm 2** Deploying stage (system exploitation by the user)

1: The user generates the user-level task requirement $s_0^{\text{user}}$ based on the unlabeled data $D_0^u$ and limited labeled data $D_0^l$ defined on $\mathcal{X}_0^{\text{user}} \times \mathcal{Y}$.
2: The user submits the user-level requirement $s_0^{\text{user}}$ to the learnware dock system.
3: The learnware dock system uses the projection functions $\{h_i\}_i^Q$ to generate the system-level requirement $s_0$.
4: The learnware dock system recommends one learnware $l_i$ based on the system-level requirement $s_0$ and the system-level specifications $\{s_i\}_{i=1}^N$ and provides the toolkit $\{g_i, h_i\}_i^Q$ to the user.
5: The user reuses recommended learnware $l_i$ with the toolkit $\{g_i, h_i\}_i^Q$ on the task.

---

### D.2  Subspace learning and system-level specification generation

After the developer-level specification $s^{\text{dev}}$ generation, the developer submits the model $f$ with specification $s^{\text{dev}}$ to the learnware dock system. The system exploits the relationship of different feature spaces and manages heterogeneous models by assigning system-level specification $s^{\text{sys}}$. The details of subspace learning and system-level specification generation are described in Algorithm 3.

## E  Optimization

### E.1  Optimization of specification generation specialized for classification tasks

The RKME$_L$ specification represented by $R_L = (\boldsymbol{\beta}, Z, Y) = \{(\beta_m, z_m, y_m)\}_{m=1}^M$ sketches both the marginal distribution $P_X$ of the training data and the conditional distribution $P_{X|Y}$ of the model prediction, which is obtained through minimizing the following objective over $R_L$:

$$\left\| \sum_{n=1}^N \frac{1}{N} k\left(\boldsymbol{x}_n, \cdot\right) - \sum_{m=1}^M \beta_m k\left(\boldsymbol{z}_m, \cdot\right) \right\|_{\mathcal{H}_k}^2 + \theta \sum_{c=1}^C \left\| \sum_{n \in \mathcal{I}_c} \frac{1}{N} k\left(\boldsymbol{x}_n, \cdot\right) - \sum_{m \in \mathcal{I}_c'} \beta_m k\left(\boldsymbol{z}_m, \cdot\right) \right\|_{\mathcal{H}_k}^2, \quad (5)$$

This objective consists of two parts: the first term sketches the marginal distribution $P_X$ of the training data, while the second term sketches the conditional distribution $P_{X|Y}$ of the model prediction. This objective can be optimized by iterative optimization, which is detailed in the following.

---
**Algorithm 3** System-level specification generation
---
**Input**: All developer-level specifications $\{s_i^{\text{dev}}\}_{i=1}^N$.
**Hyper-Parameters**: batch size $B$, temperature $\tau$, corruption rate $c$, max iteration $T$, trade-off parameters $\alpha_1, \alpha_2, \alpha_3, \alpha_4$
**Output**: System-level specifications $\{s_i^{\text{sys}}\}_{i=1}^N$, learnware dock system toolkit $\{t_i := (h_i, g_i)\}_{i=1}^Q$, system engine $F(\cdot)$

1: **while** Max epoch is not achieved **do**
2:     Initialize the system engine $F(\cdot)$ and dock system toolkit $\{t_i := (h_i, g_i)\}_{i=1}^Q$.
3:     **for** each specification $s_i^{\text{dev}}$ in all developer-level specifications $\{s_i^{\text{dev}}\}_{i=1}^N$ **do**
4:        **for** sampled mini-batch $\{\beta_j, z_j := (x_j, y_j)\}_{j=1}^B$ **do**
5:           split $x_j$ according to the feature blocks $\{\mathcal{X}_i\}_{i=1}^Q$: $\{x_j^{(k)}\}_{k \in C}$.
6:           let $v_j^{(k)} = h_i(x_j^{(k)})$.      # embeddings of each blocks.
7:           define $\mathcal{L}_{\text{cont}} := -\sum_{i=1}^B \beta_i \sum_{k=1}^K \sum_{k' \neq k}^K \log \frac{\exp \psi\left(v_i^k, v_i^{k'}\right)}{\sum_{j=1}^B \sum_{k^\dagger=1}^K \exp \psi\left(v_i^k, v_j^{k^\dagger}\right)}$ where $\psi$ is the cosine similarity function.
8:           let $\hat{x}_j^{(k)} = g_i(v_j^{(k)})$ and concatenate them as $\hat{x}_j$. # reconstructed samples.
9:           define $\mathcal{L}_{\text{reconstructed}} = \sum_{J=1}^B \sum_{k \in C} \beta_j \|x_j^{(k)} - \hat{x}_j^{(k)}\|^2$
10:           let $\hat{y}_j = F(v_j)$ where $v_j = \text{mean}(\{v_j^{(k)}\}_{k \in C})$ # simple classifier prediction.
11:           define $\mathcal{L}_{\text{supervised}} = \sum_{j=1}^B \beta_j l(y_j, \hat{y}_j)$ where $l$ is cross entropy for classification problem and mean squared error for regression problem.
12:           $\mathcal{L} = \mathcal{L} + \alpha_1 \mathcal{L}_{\text{reconstruct}} + \alpha_2 \mathcal{L}_{\text{contrastive}} + \alpha_3 \mathcal{L}_{\text{supervised}}$
13:           update the encoder networks $\{h_i\}_{i=1}^Q$, decoder networks $\{g_i\}_{i=1}^Q$ and system engine $F(\cdot)$ according to $\mathcal{L}$ using gradient descent.
14:        **end for**
15:     **end for**
16: **end while**
17: generate system-level specification $s_i^{\text{sys}} := \{\beta_j, (v_j, y_j)\}_{j=1}^{m_i}$ based on the developer-level specification $s_i^{\text{dev}} := \{\beta_j, (x_j, y_j)\}_{j=1}^{m_i}$ by replacing the raw sample $x_j$ with the embedding $v_j$.
18: **return** System-level specifications $\{s_i\}_{i=1}^N$ with corresponding toolkit $\{t_i := (h_i, g_i)\}_{i=1}^Q$ consists of encoder networks and decoder networks, system engine $F(\cdot)$.
---

Denote $\boldsymbol{\beta} = (\beta_1, \cdots, \beta_M)$, $Z = \{z_1, \cdots, z_M\}$ and $Y = \{y_1, \cdots, y_M\}$, expanding Eq. (5) gives

$$F(\boldsymbol{\beta}, Z, Y) = \sum_{n,m=1}^N \frac{1}{N^2} k(x_n, x_m) + \sum_{n,m=1}^M \beta_n \beta_m k(z_n, z_m) - 2 \sum_{n=1}^N \sum_{m=1}^M \frac{\beta_m}{N} k(x_n, z_m)$$

$$+ \theta \sum_{c=1}^C \left( \sum_{n,m \in \mathcal{I}_c} \frac{1}{N^2} k(x_n, x_m) + \sum_{n,m \in \mathcal{I}_c'} \beta_n \beta_m k(z_n, z_m) - 2 \sum_{n \in \mathcal{I}_c} \sum_{m \in \mathcal{I}_c'} \frac{\beta_m}{N} k(x_n, z_m) \right).$$

The distance $F(\boldsymbol{\beta}, Z, Y)$ can also be rewritten as

$$\alpha^T K_{xx} \alpha + \beta^T K_{zz} \beta - 2\alpha^T K_{xz} \beta + \theta \sum_{c=1}^C \left( \alpha_c^T K_{x_c x_c} \alpha_c + \beta_c^T K_{z_c z_c} \beta_c - 2\alpha_c^T K_{x_c z_c} \beta_c \right)$$

$$= \alpha^T (K_{xx} + \theta K_{xx}^b) \alpha + \beta^T (K_{zz} + \theta K_{zz}^b) \beta - 2\alpha^T (K_{xz} + \theta K_{xz}^b) \beta$$

$$= \alpha^T K_{xx}' \alpha + \beta^T K_{zz}' \beta - 2\alpha^T K_{xz}' \beta,$$

where $\alpha$ is the vector containing $N$ elements of 1.

Next, we address the optimization of Eq. (5). First, we generate $Y$ while preserving the class ratio of the original labels $\{y_n\}_{n=1}^N$. Then, we proceed to optimize $\boldsymbol{\beta}$ and $Z$.

**Fix $Z$ and update $\beta$.** Suppose vectors in $Z$ are fixed, setting $\frac{\partial F(\beta, Z)}{\partial \beta} = 0$ obtains the closed-form solution of $\beta$ using pseudo-inverse of $K'_{zz}$ :

$$\beta = (K'_{zz})^\dagger K'_{zx} \alpha.$$

**Fix $\beta$ and update $Z$.** When $\beta$ is fixed, $\{z_1, \cdots, z_M\}$ in $Z$ are independent in Eq. (5), therefore we can iteratively run gradient descent on each $z_m$ as

$$\boldsymbol{z}_m^{(t)} = \boldsymbol{z}_m^{(t-1)} - \eta \frac{\partial F(\beta, Z)}{\partial \boldsymbol{z}_m}.$$

We first review the gradient $\frac{\partial G(\beta, Z)}{\partial \boldsymbol{z}_m}$:

$$\frac{\partial G(\beta, Z)}{\partial \boldsymbol{z}_m} = 2 \sum_{n=1}^{M} \beta_n \beta_m \frac{\partial k(\boldsymbol{z}_n, \boldsymbol{z}_m)}{\partial \boldsymbol{z}_m} - 2 \sum_{n=1}^{N} \frac{\beta_m}{N} \frac{\partial k(\boldsymbol{x}_n, \boldsymbol{z}_m)}{\partial \boldsymbol{z}_m}$$

$$= 2 \sum_{n=1}^{M} \beta_n \beta_m (-2\gamma k(\boldsymbol{z}_n, \boldsymbol{z}_m)(\boldsymbol{z}_m - \boldsymbol{z}_n)) - 2 \sum_{n=1}^{N} \frac{\beta_m}{N} (-2\gamma k(\boldsymbol{x}_n, \boldsymbol{z}_m)(\boldsymbol{z}_m - \boldsymbol{x}_n))$$

$$= -4\gamma \beta_m (\sum_{n=1}^{M} \beta_n k(\boldsymbol{z}_n, \boldsymbol{z}_m)(\boldsymbol{z}_m - \boldsymbol{z}_n) - \frac{1}{N} \sum_{n=1}^{N} k(\boldsymbol{x}_n, \boldsymbol{z}_m)(\boldsymbol{z}_m - \boldsymbol{x}_n))$$

then, we consider the gradient in our problem $\frac{\partial F(\beta, Z)}{\partial \boldsymbol{z}_m}$, which is calculated as follows:

$$2 \sum_{n=1}^{M} \beta_n \beta_m \frac{\partial k(\boldsymbol{z}_n, \boldsymbol{z}_m)}{\partial \boldsymbol{z}_m} - 2 \sum_{n=1}^{N} \frac{\beta_m}{N} \frac{\partial k(\boldsymbol{x}_n, \boldsymbol{z}_m)}{\partial \boldsymbol{z}_m}$$

$$+ \theta \left( 2 \sum_{n \in \mathcal{I}'_c} \beta_n \beta_m \frac{\partial k(\boldsymbol{z}_n, \boldsymbol{z}_m)}{\partial \boldsymbol{z}_m} - 2 \sum_{n \in \mathcal{I}_c} \frac{\beta_m}{N} \frac{\partial k(\boldsymbol{x}_n, \boldsymbol{z}_m)}{\partial \boldsymbol{z}_m} \right)$$

$$= 2 \sum_{n=1}^{M} \beta_n \beta_m (-2\gamma k(\boldsymbol{z}_n, \boldsymbol{z}_m)(\boldsymbol{z}_m - \boldsymbol{z}_n)) - 2 \sum_{n=1}^{N} \frac{\beta_m}{N} (-2\gamma k(\boldsymbol{x}_n, \boldsymbol{z}_m)(\boldsymbol{z}_m - \boldsymbol{x}_n))$$

$$+ \theta \left( 2 \sum_{n \in \mathcal{I}'_c} \beta_n \beta_m (-2\gamma k(\boldsymbol{z}_n, \boldsymbol{z}_m)(\boldsymbol{z}_m - \boldsymbol{z}_n)) - 2 \sum_{n \in \mathcal{I}_c} \frac{\beta_m}{N} (-2\gamma k(\boldsymbol{x}_n, \boldsymbol{z}_m)(\boldsymbol{z}_m - \boldsymbol{x}_n)) \right)$$

$$= -4\gamma \beta_m \left( \sum_{n=1}^{M} \beta_n k(\boldsymbol{z}_n, \boldsymbol{z}_m)(\boldsymbol{z}_m - \boldsymbol{z}_n) - \frac{1}{N} \sum_{n=1}^{N} k(\boldsymbol{x}_n, \boldsymbol{z}_m)(\boldsymbol{z}_m - \boldsymbol{x}_n) \right.$$

$$+ \theta \left( \sum_{n \in \mathcal{I}'_c} \beta_n k(z_n, z_m)(z_m - z_n) - \frac{1}{N} \sum_{n \in \mathcal{I}_c} k(x_n, z_m)(z_m - x_n) \right) \Bigg)$$

$$= -4\gamma \beta_m \left( \sum_{n=1}^{M} \beta_n (1 + \theta \mathbb{I}(y_{z_n} = c)) k(\boldsymbol{z}_n, \boldsymbol{z}_m)(\boldsymbol{z}_m - \boldsymbol{z}_n) \right.$$

$$\left. - \frac{1 + \theta \mathbb{I}(y_{\boldsymbol{x}_n} = c)}{N} \sum_{n=1}^{N} k(\boldsymbol{x}_n, \boldsymbol{z}_m)(\boldsymbol{z}_m - \boldsymbol{x}_n) \right),$$

where $\mathcal{I}_c$ is the sample indices of class c of sample $\boldsymbol{z}_m$.

### E.2 Optimization of user requirement generation

When the user has major unlabeled data $D^u = \{\boldsymbol{x}_i\}_{i=1}^{N_l}$ and limited labeled data $D^l = \{(\tilde{\boldsymbol{x}}_i, y_i)\}_{i=1}^{N_u}$ (We adjust the notation slightly for a clearer description of the optimization process), we can also

find a weighted labeled set of points $R = (\boldsymbol{\beta}, Z, Y) = \{(\beta_m, \boldsymbol{z}_m, y_m)\}_{m=1}^{M}$ to sketch the user data as task requirement.

The weighted labeled set can be generated by minimizing the following distance:

$$\left\| \sum_{n=1}^{N_u} \frac{1}{N_u} k(x_n, \cdot) - \sum_{m=1}^{M} \beta_m k(z_m, \cdot) \right\|_{\mathcal{H}_k}^2 + \theta \sum_{c=1}^{C} \left\| \sum_{n \in \mathcal{I}_c} \frac{1}{N_l} k(\tilde{x}_n, \cdot) - \sum_{m \in \mathcal{I}_c'} \beta_m k(z_m, \cdot) \right\|_{\mathcal{H}_k}^2, \quad (6)$$

which can be optimized by iterative optimization. The details are described as follows.

Denote $\boldsymbol{\beta} = (\beta_1, \cdots, \beta_M)$, $Z = \{\boldsymbol{z}_1, \cdots, \boldsymbol{z}_M\}$ and $Y = \{y_1, \cdots, y_M\}$, expanding Eq. (6) gives

$$F(\boldsymbol{\beta}, Z, Y) = \sum_{n,m=1}^{N_u} \frac{1}{N_u^2} k(\boldsymbol{x}_n, \boldsymbol{x}_m) + \sum_{n,m=1}^{M} \beta_n \beta_m k(\boldsymbol{z}_n, \boldsymbol{z}_m) - 2 \sum_{n=1}^{N_u} \sum_{m=1}^{M} \frac{\beta_m}{N_u} k(\boldsymbol{x}_n, \boldsymbol{z}_m)$$

$$+ \theta \sum_{c=1}^{C} \left( \sum_{n,m \in \mathcal{I}_c} \frac{1}{N_l^2} k(\tilde{\boldsymbol{x}}_n, \tilde{\boldsymbol{x}}_m) + \sum_{n,m \in \mathcal{I}_c'} \beta_n \beta_m k(\boldsymbol{z}_n, \boldsymbol{z}_m) - 2 \sum_{n \in \mathcal{I}_c} \sum_{m \in \mathcal{I}_c'} \frac{\beta_m}{N_l} k(\tilde{\boldsymbol{x}}_n, \boldsymbol{z}_m) \right)$$

The distance $F(\boldsymbol{\beta}, Z, Y)$ can also be rewritten as

$$\alpha^T K_{xx} \alpha + \beta^T K_{zz} \beta - 2\alpha^T K_{xz} \beta + \theta \sum_{c=1}^{C} \left( \tilde{\alpha}_c^T K_{\tilde{\boldsymbol{x}}_c \tilde{\boldsymbol{x}}_c} \tilde{\alpha}_c + \beta_c^T K_{z_c z_c} \beta_c - 2\tilde{\alpha}_c^T K_{\tilde{\boldsymbol{x}}_c z_c} \beta_c \right)$$

$$= \left( \alpha^T K_{xx} \alpha + \theta \tilde{\alpha}^T K_{\tilde{\boldsymbol{x}}\tilde{\boldsymbol{x}}}^b \tilde{\alpha} \right) + \beta^T (K_{zz} + \theta K_{zz}^b) \beta - 2(\alpha^T K_{xz} + \theta \tilde{\alpha}^T K_{\tilde{\boldsymbol{x}}z}^b) \beta$$

$$= \left( \alpha^T K_{xx} \alpha + \theta \tilde{\alpha}^T K_{\tilde{\boldsymbol{x}}\tilde{\boldsymbol{x}}}^b \tilde{\alpha} \right) + \beta^T (K_{zz}') \beta - 2(\alpha^T K_{xz} + \theta \tilde{\alpha}^T K_{\tilde{\boldsymbol{x}}z}^b) \beta$$

where $\alpha$ is the vector containing $N_u$ elements of 1, $\tilde{\alpha}$ is the vector containing $N_l$ elements of 1.

Then, we discuss the optimization of Eq. (6). We first generate $Y$, which will keep the class ratio of $\{y_n\}_{n=1}^{N}$. Then we optimize $\beta, Z$.

**Fix $Z$ and update $\beta$.** Suppose vectors in $Z$ are fixed, setting $\frac{\partial F(\boldsymbol{\beta}, Z)}{\partial \boldsymbol{\beta}} = 0$ obtains the closed-form solution of $\boldsymbol{\beta}$ using pseudo-inverse of $K_{zz}'$ :

$$\boldsymbol{\beta} = (K_{zz}')^\dagger (K_{zx} \alpha + \theta K_{z\tilde{\boldsymbol{x}}}^b \tilde{\alpha}).$$

**Fix $\beta$ and update $Z$.** When $\boldsymbol{\beta}$ is fixed, $\{z_1, \cdots, z_M\}$ in $Z$ are independent in Eq. (6), therefore we can iteratively run gradient descent on each $z_m$ as

$$\boldsymbol{z}_m^{(t)} = \boldsymbol{z}_m^{(t-1)} - \eta \frac{\partial F(\boldsymbol{\beta}, Z)}{\partial \boldsymbol{z}_m}.$$

The gradient in our problem $\frac{\partial F(\boldsymbol{\beta}, Z)}{\partial \boldsymbol{z}_m}$ can be calculated as:

$$\frac{\partial F(\boldsymbol{\beta}, Z)}{\partial \boldsymbol{z}_m} = 2 \sum_{n=1}^{M} \beta_n \beta_m \frac{\partial k(\boldsymbol{z}_n, \boldsymbol{z}_m)}{\partial \boldsymbol{z}_m} - 2 \sum_{n=1}^{N_l} \frac{\beta_m}{N_l} \frac{\partial k(\boldsymbol{x}_n, \boldsymbol{z}_m)}{\partial \boldsymbol{z}_m}$$

$$= -4\gamma \beta_m \left( \sum_{n=1}^{M} \beta_n (1 + \theta \mathbb{I}(y_{\boldsymbol{z}_n} = c)) k(\boldsymbol{z}_n, \boldsymbol{z}_m)(\boldsymbol{z}_m - \boldsymbol{z}_n) \right.$$

$$\left. - \frac{1}{N_u} \sum_{n=1}^{N_u} k(\boldsymbol{x}_n, \boldsymbol{z}_m)(\boldsymbol{z}_m - \boldsymbol{x}_n) - \frac{\theta}{N_l} \sum_{n \in \mathcal{I}_c} k(\tilde{\boldsymbol{x}}_n, \boldsymbol{z}_m)(\boldsymbol{z}_m - \tilde{\boldsymbol{x}}_n) \right)$$

where $\mathcal{I}_c$ represents the indices of labeled samples $\tilde{\boldsymbol{x}}_n$ belonging to the same class as $\boldsymbol{z}_m$.

### E.3  Summary

The generation mechanism for model specification and user requirement is summarized in Figure 6.

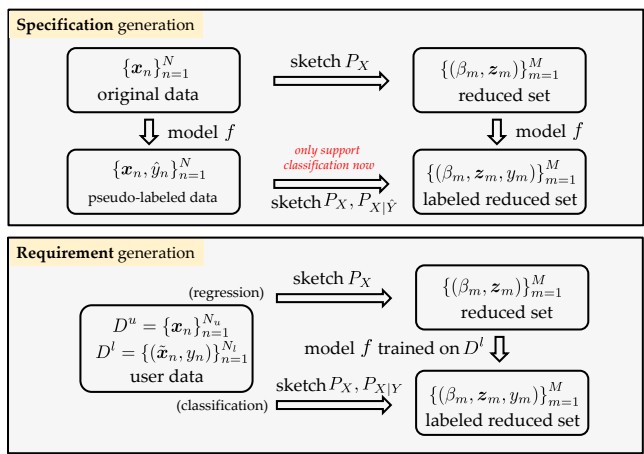

Figure 6: summarized mechanisms for $\text{RKME}_\text{L}$ generation.

# F  Experiments

## F.1  More details for basic information

**Dataset details.**  The basic information of used datasets is summarized in Table 4 and Table 5.

Table 4: Details for classification tasks

| Dataset name | #classes | #features | #instances |
|---|---|---|---|
| openml__credit-g__31 | 1 | 20 | 1000 |
| openml__semeion__9964 | 10 | 256 | 1593 |
| openml__mfeat-karhunen__16 | 10 | 64 | 2000 |
| openml__splice__45 | 3 | 60 | 3190 |
| openml__gina_agnostic__3891 | 1 | 970 | 3468 |
| openml__Bioresponse__9910 | 1 | 1776 | 3751 |
| openml__sylvine__168912 | 1 | 20 | 5124 |
| openml__christine__168908 | 1 | 1636 | 5418 |
| openml__first-order-theorem-proving__9985 | 6 | 51 | 6118 |
| openml__satimage__2074 | 6 | 36 | 6430 |
| openml__fabert__168910 | 7 | 800 | 8237 |
| openml__GesturePhaseSegmentationProcessed__14969 | 5 | 32 | 9873 |
| openml__robert__168332 | 10 | 7200 | 10000 |
| openml__artificial-characters__14964 | 10 | 7 | 10218 |
| openml__eye_movements__3897 | 3 | 27 | 10936 |
| openml__nursery__9892 | 4 | 8 | 12958 |
| openml__eeg-eye-state__14951 | 1 | 14 | 14980 |
| openml__magic__146206 | 1 | 10 | 19020 |
| openml__riccardo__168338 | 1 | 4296 | 20000 |
| openml__guillermo__168337 | 1 | 4296 | 20000 |
| openml__nomao__9977 | 1 | 118 | 34465 |
| openml__Click_prediction_small__190408 | 1 | 11 | 39948 |
| openml__volkert__168331 | 10 | 180 | 58310 |

**Experiment configuration.**  Each dataset is split into training and test sets with a 4:1 ratio [McElfresh et al., 2023]. The output of regression tasks is scaled to [0,1]. The feature space is randomly divided into four equal blocks. We create four feature spaces for developer tasks from all three-block combinations and six feature spaces for user tasks from all two-block combinations. Our encoder, decoder, and system classifier are two-layer ResNets [He et al., 2016] for tabular data, with subspace and hidden layer dimensions set to 16 and 32, respectively. We optimize using Adam [Kingma and Ba, 2015]. For user tasks, we sample 100 labeled data points from the training set, using stratified

Table 5: Details for regression tasks

| Dataset name | #classes | #features | #instances |
|---|---|---|---|
| openml__pbc__4850 | 1 | 19 | 418 |
| openml__colleges__359942 | 1 | 44 | 7063 |
| openml__cpu_small__4883 | 1 | 12 | 8192 |
| openml__kin8nm__2280 | 1 | 8 | 8192 |
| openml__dataset_sales__190418 | 1 | 14 | 10738 |
| openml__california__361089 | 1 | 8 | 20640 |
| openml__aloi__12732 | 1 | 128 | 108000 |

sampling for classification and binning for regression. The coefficients for contrastive, reconstruction and supervised losses are set to 100, 1, and 1, respectively. Developers or users train the model using LightGBM [Ke et al., 2017] with grid search. All experiments are repeated five times.

**Model training details.** Tree-based models provide strong performance and high efficiency for supervised tabular tasks, so we use the popular efficient tree ensemble method LightGBM [Ke et al., 2017] to train developer models and include it as a comparative method. The hyper-parameter search space used for the developer and user LightGBM models consists of a list of specific combinations over parameters `learning_rate`, `num_leaves` and `max_depth`: (0.015, 224, 66), (0.005, 300, 50), (0.01, 128, 80), (0.15, 224, 80), and (0.01, 300, 66).

**Baseline details.** More details of deep tabular network are described as follows:

- `TabPFN`: For TabPFN, we use the official checkpoint as a pre-trained model and further fine-tune it on downstream datasets. When testing TabPFN on datasets with more than 1,000 instances or 100 features, we randomly sample up to 1,000 instances and 100 features from the full training set, repeating this process three times. The final output is obtained by averaging the predictions from these trials.
- `Xtab`: We utilize the pre-trained backbone with the most iterations from XTab's official implementation. Since this model is based on the FT-Transformer from AutoGluon, which currently does not support fine-tuning on tasks with more than 300 features, we generate random subsets of up to 300 features from the overall training dataset, repeating this procedure three times. The final prediction is derived by averaging the results across three evaluations on the target datasets, using XTab's lightweight fine-tuning approach over 15 epochs.
- `Transtab`: We employ the official version of TransTab v0.0.5, testing it in a contrastive learning setting. This involves initial contrastive pre-training on the developers' training data, followed by fine-tuning on user tasks. We adopt a supervised contrastive learning objective during pre-training, which has been shown to outperform the unsupervised version as noted in [Wang and Sun, 2022]. For regression tasks, a linear regressor is integrated during the fine-tuning phase. To manage memory constraints with large datasets, we subsample the developers' training sets to 20,000 data points and 100 features.

**Computation resources.** Experiments were conducted using a Tesla A100 80GB GPU, two Intel Xeon Platinum 8358 CPUs with 32 cores each (base clock 2.6 GHz, turbo boost 3.4 GHz), and 512 GB of RAM.

## F.2 All user curves for classification tasks

The all user performance curves for classification tasks are shown in Figure 7.

## F.3 Ablation study

**Ablation study for subspace learning loss by adding loss in sequence.** We undertake an ablation study to evaluate the efficacy of our subspace learning procedure, as depicted in Tables 6 and 7. Initially, we utilize only the contrastive loss as the baseline. This helps in learning encoding functions for different feature blocks, ensuring consistency in embeddings across all sample slices. However, the reconstruction functions learned in this manner are subpar. By adding a reconstruction loss,

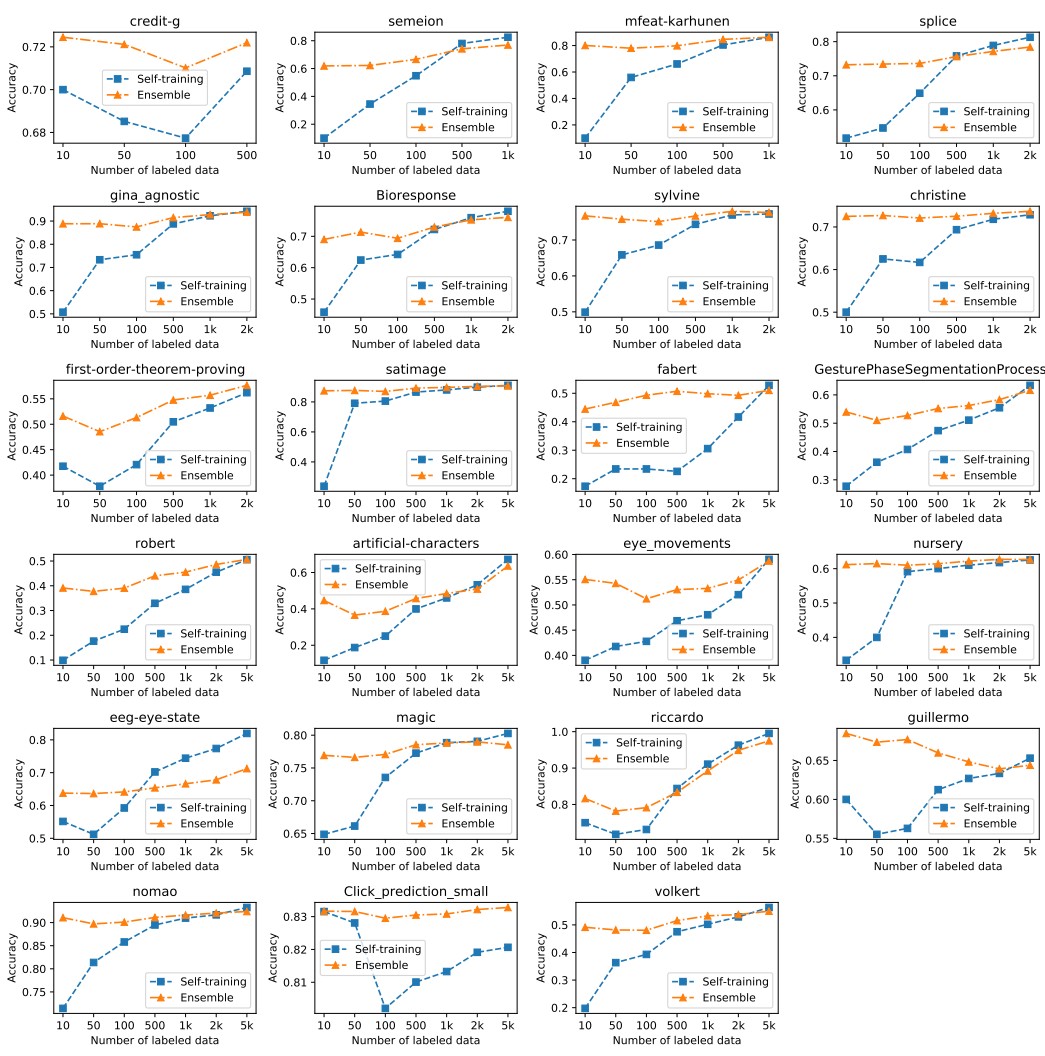

Figure 7: All user performance curve for classification tasks.

the performance of decoding functions is matched with the encoding functions. Despite the initial effectiveness of unsupervised subspace learning, further enhancements are achieved by incorporating label information through our proposed specification. With the addition of supervised loss, we ultimately attain optimal performance for both classification and regression tasks.

Table 6: Ablation study for subspace learning loss by *adding loss in sequence* for classification tasks.

| | contrastive | + reconstruction | + supervised |
|---|---|---|---|
| acc(%) | 61.1 | 66.2 | **67.4** |

Table 7: Ablation study for subspace learning loss by *adding loss in sequence* for regression tasks.

| | contrastive | + reconstruction | + supervised |
|---|---|---|---|
| rmse | 0.208 | 0.200 | **0.194** |

### F.4 Code availability

The code can be found at `https://github.com/LAMDA-TP/Hetero-Learnware-Label-Info`.

