# OpenReview forum: "Handling Learnwares from Heterogeneous Feature Spaces with Explicit Label Exploitation"
_NeurIPS.cc/2024/Conference — NeurIPS 2024 poster_

### Official Review · Reviewer_HLyF · 2024-07-03

**Soundness:** 2
**Presentation:** 3
**Contribution:** 3
**Rating:** 5
**Confidence:** 3

**Summary:**

The paper designs a learnware system to handle the heterogeneous feature spaces. It applies the label information to manage the heterogeneous models. Then it also proposes a strategy to match the model with the additional conditional distribution. The paper also conducts experiments to compare with other learnware systems.

**Strengths:**

1. The observation that matching with only the marginal distribution is not enough is interesting and novel. The paper improves the matching with the conditional distribution to tackle this problem.

2. The experimental results are good.

3. The paper is well presented and easy to follow.

**Weaknesses:**

1. The paper focuses on the heterogeneous feature spaces in learnwares. However, it seems that the paper uses subspace learning to handle the heterogeneous feature spaces. Subspace learning is a standard and commonly used method to handle heterogeneous features. Therefore, the contributions to handle heterogeneous features of this paper seem limited. The proposed technology, such as to use the conditional distribution, seems to be irrelative to handling the heterogeneous features because it can also be used in a homogeneous setting. The paper should clarify the relations between each technical contribution and the heterogeneous feature spaces.

2. In the experiments, the paper should compare with more SOTA methods which handle the heterogeneous features. In current version, most compared methods are not designed for heterogeneous features.

3. In the experiments, the heterogeneous feature spaces are randomly divided from the original data sets. However, in real applications, the distribution of heterogenous features is not in this case. For example, in real-world applications, each feature space has its own semantics and is not random. Therefore, it would be better to conduct experiments on at least one real-world heterogeneous data.

**Questions:**

In Eq.(2), there is a term $L_{similar}$,  but in Section 5.2, there is a term called $L_{contrastive}$. Are them the same? Is this a typo or are they two different losses?

**Limitations:**

The paper claims the limitations in Checklist.

---

> ### Author Rebuttal · Authors · 2024-08-06
>
> **Q1: Clarify the relations between each technical contribution and the heterogeneous feature spaces.**
>
> Thank you for your comments. We will provide a more detailed discussion in the revised version. The following is a brief explanation:
>
> **Subspace learning is a standard method for handling heterogeneous features. The critical aspects are from what to learn and how to learn it.** The learnware framework protects model providers' privacy without accessing their raw data and avoids collecting auxiliary data across feature spaces. **Without raw and auxiliary data, how can the learnware dock system effectively learn a unified subspace?**
>
> In the learnware paradigm, subspace learning can be performed based on the RKME specification of the model. The RKME specification compresses the marginal distribution $P(X)$ through a small number of weighted sample points. However, **previous methods that lack label information rely on unsupervised learning techniques, leading to suboptimal subspace learning results**. Without label information, categories from different tasks may overlap in the subspace, impacting model recommendation and reuse effectiveness.
>
> To address this, **we introduce supervised information into subspace learning to better align samples from each class of different tasks within the subspace**, enhancing the heterogeneous model hub's performance. Thus, we **extend the original RKME specification representation $\\{\beta\_m,\boldsymbol{z}\_m\\}\_{m=1}^M$ to include label information**, obtaining $\text{RKME}\_\text{L}$, represented as $\\{\beta\_m,\boldsymbol{z}\_m, y\_m\\}\_{m=1}^M$. We propose two methods: compressing the marginal distribution first and then generating the label, or **simultaneously compressing both the marginal distribution $P(X)$ and the conditional distribution $P(Y|X)$**. By incorporating label information, we **add supervised term for subspace learning**, obtaining a subspace with better properties and corresponding mapping functions to original feature spaces.
>
> In summary, our main techniques are designed to enhance subspace learning.
>
> ---
>
> **Q2: In the experiments, the paper should compare with more SOTA methods which handle the heterogeneous features. In current version, most compared methods are not designed for heterogeneous features.**
>
> Thank you for your comments. **Indeed, most of the compared methods are designed for heterogeneous features and are SOTA methods.** We will describe the contenders more clearly in the revised version.
>
> **This paper explores using knowledge from tasks with heterogeneous feature spaces to improve user task performance.**
>
> First, we evaluate user self-training performance using LightGBM with the same training setup as the learnware preparation for fairness.
>
> Next, **we compare our approach with recent SOTA methods that use the heterogeneous model hub for the user task**:
>
> - **Align_unlabel (KDD 2024):** Minimizes MMD distance between the user task and learnware task based on RKME. We modified it to traverse the best model in the hub, improving performance.
> - **Align_label (KDD 2024):** Builds on Align_unlabel by using user-labeled data for feature augmentation, further improving performance. We also traverse the hub for the best model.
> - **Hetero (IJCAI 2023):** Generates a subspace for model recommendation/reuse using basic RKME without label information, creating a linear mapping function.
>
> These solutions learn feature space mapping without label information. Our paper fully utilizes label information, resulting in significantly better performance.
>
> **We also compare with other SOTA methods that use raw heterogeneous task data, compromising privacy**:
>
> - **TabPFN (ICLR 2023):** Uses a Transformer to fit the posterior predictive distribution from synthetic tasks and applies it to real-world tasks. It outperforms other deep methods in the small data paradigm.
> - **Transtab (NeurIPS 2022):** Uses feature descriptions and values to generate a unified subspace and trains a shared Transformer backbone, pioneering deep network training across heterogeneous tasks.
> - **Xtab (ICML 2023):** Trains a shared backbone for different tasks with a specific input processor and prediction head, without using feature descriptions, offering flexibility in real-world scenarios.
>
> **In summary, the contenders are highly relevant to heterogeneous feature spaces. We compared not only with SOTA methods that exploit the heterogeneous model hub but also with other SOTA methods that use raw data from heterogeneous tasks for new user tasks.**
>
> ---
>
>
> **Q3: Conduct experiments on at least one real-world heterogeneous data.**
>
> Thanks for your suggestion, we add **two real-world projects** to demonstrate the efficacy of proposed method, please see author rebuttal for details.
>
> ---
>
> **Q4: In Eq.(2), there is a term Lsimilar, but in Section 5.2, there is a term called Lcontrastive. Are them the same? Is this a typo or are they two different losses?**
>
> We apologize for any confusion caused. Thanks for your careful reading.
>
> **In short, $L\_{\text{similar}}$ is a general term for similarity loss, and $L\_{\text{contrastive}}$ is a specific implementation.**
>
> In Section 4.2, we define the general objective for subspace learning as Equation (2), based on all model specifications $\\{\boldsymbol{s}\_i^{\text{dev}}:=\\{(\boldsymbol{\beta}\_{ij},\boldsymbol{z}\_{ij},y\_{ij})\\}\_{j=1}^{m\_i}\\}\_{i=1}^N$:
> $$\min\_{\\{h\_k,g\_k\\}\_{k=1}^Q} L=\alpha\_1L\_{\text{reconstruction}} + \alpha\_2L\_{\text{similar}} + \alpha\_3L\_{\text{supervised}}.$$
> The similarity loss ensures that embeddings of different slices of the sample $\boldsymbol{z}_j$ are similar. This can be implemented in various ways, such as through contrastive loss, manifold loss, etc.
>
> In Section 5.2, we provide a detailed implementation of the loss function, specifically using contrastive loss to measure similarity, which is a quite popular loss for self-supervised learning.

---

> > ### Comment · Reviewer_HLyF · 2024-08-12
> >
> > Thanks for the authors' responses. The authors have addressed some of my concerns. Since I'm not quite familiar with the learnwares for heterogeneous features (as shown in my Confidence), I'm not sure about the contribution to this community. Especially Reviewer PBMH also proposes some questions about the contributions.

---

> > > ### Author Response · Authors · 2024-08-12
> > >
> > > Thank you for reviewing our manuscript. Please feel free to let us know your other concerns. We are ready to provide further explanations to address any issues you may have. The following is a brief discussion of our paper contribution:
> > >
> > > The learnware paradigm allows users to reuse existing models instead of training from scratch. Expanding its use to heterogeneous feature spaces can broaden its applications, but current approaches either compromise privacy (MLJ 2022) or are less effective in subspace learning (MLJ2022, IJCAI 2023).
> > >
> > > **Our paper is the first to utilize label information to enhance handling heterogeneous learnwares, marking a significant advancement of handling heterogeneous feature spaces from unsupervised to supervised.** We have refined the specification used for subspace learning to incorporate label information, which first encodes the model's capabilities. This new specification also facilitates better task matching between the learnware and the user task by considering the model’s abilities in the matching process, which is quite superior to previous methods based on solely the marginal distribution.
> > >
> > > Broader Impact:  As the specification is the most fundamental part of the learnware paradigm, its improvement can enhance the entire framework. **The previous specification lacked supervision information, but our new one includes it. This make top-layer procedure based on specification from unsupervised to supervised, which is a significant advancement.**  With the more powerful specification, the learnware paradigm can address more difficult problems effectively. For example, the inclusion of supervision information in the newly proposed specification will be crucial in tackling challenges such as simultaneously heterogeneous feature and label spaces. This enhanced specification also opens the possibility for applying the learnware paradigm to more complex and diverse heterogeneous scenarios.

---

### Official Review · Reviewer_PBMH · 2024-07-10

**Soundness:** 3
**Presentation:** 3
**Contribution:** 3
**Rating:** 5
**Confidence:** 3

**Summary:**

In this submission, authors focus on the learnware paradigm, which aims to help users leverage numerous existing high-performing models instead of starting from scratch. They find that label information, including model prediction and user’s minor labeled data, is crucial and previously unexplored and then explicitly explore this information to address the problem in handling learnwares from heterogeneous feature spaces.

**Strengths:**

1. Learnware is useful learning paradigm and handling learnwares from heterogeneous feature spaces has more wide application scenarios.

2. To my knowledge, for handling learnwares from heterogeneous feature spaces, it might be the first attempt towards explicitly exploiting label information, including model prediction and user's minor labeled data.

3. For this new setting, comparative studies are designed, and the experimental result validate the effectiveness of the proposed method.

**Weaknesses:**

1. To my knowledge, there have been some works aiming at handling learnwares from heterogeneous feature spaces:

[a] Peng Tan, Zhi-Hao Tan, Yuan Jiang, and Zhi-Hua Zhou. Handling learnwares developed from heterogeneous feature spaces without auxiliary data. In Proceedings of the 32nd International Joint Conference on Artificial Intelligence, pages 4235–4243, 2023.

[b] Peng Tan, Zhi-Hao Tan, Yuan Jiang, and Zhi-Hua Zhou. Towards enabling learnware to handle heterogeneous feature spaces. Machine Learning, 113(4):1839–1860, 2024.

However, these works are only cited in reference list while not adequately discussed.

2. The main contribution might be handling learnwares from heterogeneous feature spaces via explicitly exploiting label information. The current abstract does not focus on this contribution. Moreover, RKME is the most commonly-used specification in learnware, this paper is also based on RKME, so discussion on the newly proposed specification should be given in detail, especially the differences with existing RKME specification. If I misunderstand something, authors can rely in the rebuttal phase.

3. It is mentioned that "the recommended heterogeneous learnware significantly outperforms user self-training with limited labeled data", it is similar to semi-supervised learning. Is it fair to compare the proposed method with user self-training with limited labeled data?

**Questions:**

Please clarify the weaknesses mention above.

**Limitations:**

In this paper, authors discussed the limitations as follows:

This paper assumes that all models with heterogeneous feature spaces share the same label space. However, this assumption can be extended to include heterogeneous label spaces as well, through the use of multiple learnware recommendations proposed in previous work.

---

> ### Author Rebuttal · Authors · 2024-08-06
>
> **Q1: To my knowledge, there have been some works aiming at handling learnwares from heterogeneous feature spaces. However, these works are only cited in reference list while not adequately discussed.**
>
> Thanks for your careful reading. We will discuss more related papers in the revised version. Here is a brief discussion:
>
> These works focus on **constructing and utilizing heterogeneous learnware markets in specific scenarios** with relatively fixed feature combinations, such as relational databases constructed from multiple related tables. For a specific task, the complete feature space consists of several blocks, and the user and learnware feature spaces are combinations of these blocks.
>
> - **MLJ 2024:** The **first work** to consider heterogeneous learnware. **The original training data is accessible, and auxiliary data across the entire feature space is collected.** The paper assigns specifications to heterogeneous models, learns the subspace using the model's original data and auxiliary data, and then generates the RKME specification based on the projection results.
>
> - **IJCAI 2023:** This study explores **organizing and utilizing a heterogeneous model hub without accessing the original data or auxiliary data across the feature space.** It performs subspace learning via the RKME specification. Developers upload models and specifications to the learnware market. The market learns the subspace and mapping function based on the specification, aligning learnware specifications with user requirements.
>
> - **This paper:** This work **identifies the limitations of the RKME specification,** noting its inability to effectively support subspace learning and model recommendation due to the lack of label information. Unsupervised subspace learning can mix data from different tasks and categories, leading to poor model recommendation and reuse. The RKME specification only describes the marginal distribution, matching models based solely on this without considering their capabilities. **This paper embeds label information and model capabilities into the specification** to improve the heterogeneous learnware market mechanism. This **allows subspace learning to incorporate supervised information**, better aligning samples from each class of different tasks. In model recommendation, the model's classification boundary is matched with the user distribution, enhancing the performance of the heterogeneous model hub.
>
> ---
>
> **Q2: Discussion of newly proposed specifications**
>
> Thank you for your advice. We will revise the abstract and provide a detailed discussion on the newly proposed specification in the revised version. Below are the major differences between the two specifications:
>
> - **Summary:** Specifications describe a model's ability without exposing raw data. $\text{RKME}$ (TKDE 2023) sketches the marginal distribution of the training data but lacks label information, which is insufficient for handling heterogeneous learnwares. We extend $\text{RKME}$ to $\text{RKME}_\text{L}$ by adding label information to better encode the model's ability through its conditional distribution.
> - **Representation Form:** The original $\text{RKME}$ specification includes minor weighted samples $\\{\beta\_m,\boldsymbol{z}\_m\\}\_{m=1}^M$. The extended $\text{RKME}\_\text{L}$ specification includes minor *labeled* weighted samples $\\{\beta\_m,\boldsymbol{z}\_m,y\_m\\}\_{m=1}^M$, where $\beta\_m$ is the weight, $\boldsymbol{z}\_m$ is the generated sample, and $y\_m$ is the label.
> - **Generation Mechanism:** $\text{RKME}$ outlines the marginal distribution $P(X)$ of a dataset. In contrast, $\text{RKME}\_\text{L}$ includes label information, outlining both the marginal distribution $P(X)$ and the conditional distribution $P(Y|X)$. $\text{RKME}$ uses minor weighted samples to approximate the kernel mean embedding of the original data, with distance measured by MMD distance. $\text{RKME}\_\text{L}$ can either compress the marginal distribution $P(X)$ first and then generate labels or compress both $P(X)$ and $P(Y|X)$ simultaneously based on model predictions.
> - **Impact on Subspace Learning:** With $\text{RKME}$, the lack of label information can result in poor model recommendation and reuse as different task categories may overlap in the subspace. In contrast, $\text{RKME}\_\text{L}$ allows for better alignment of samples from different tasks within the same class, leading to improved performance of the heterogeneous model hub.
> - **Impact on Recommendation Mechanism:** $\text{RKME}$ only recommends models with a similar marginal distribution to the user task, ignoring the model's ability. $\text{RKME}\_\text{L}$ enables the system to match the model boundary with the user task boundary by comparing the conditional distribution $P(Y|X)$, resulting in better model recommendation outcomes.
>
> ---
>
> **Q3: It is mentioned that "the recommended heterogeneous learnware significantly outperforms user self-training with limited labeled data", it is similar to semi-supervised learning. Is it fair to compare the proposed method with user self-training with limited labeled data?**
>
> We apologize for any confusion caused and appreciate your careful reading. We will address this in the revised version.
>
> **Our problem setup aligns with supervised learning; however, the user's training data may be insufficient.** In such cases, training a model with a small expected loss is challenging due to the lack of labeled data. By leveraging the heterogeneous learnware dock system to identify and reuse existing well-trained models, users can significantly enhance task performance, as demonstrated in the paper experiments.
>
> ---
>
> **Q4: Extension for heterogeneous label space**
>
> Thanks for your suggestion. Please see author rebuttal for a brief discussion.

---

> > ### Comment · Reviewer_PBMH · 2024-08-11
> >
> > Thanks for your response. Some of my concerns have been addressed. I am open to hear discussions from other reviewers.

---

> > > ### Author Response · Authors · 2024-08-12
> > >
> > > Thank you for reviewing our manuscript. Please feel free to let us know your other concerns. We are ready to provide further explanations to address any issues you may have.

---

### Official Review · Reviewer_8wdE · 2024-07-10

**Soundness:** 3
**Presentation:** 3
**Contribution:** 3
**Rating:** 5
**Confidence:** 3

**Summary:**

This paper introduces a novel approach for utilizing heterogeneous models with diverse feature spaces in the learnware paradigm. Key innovations include incorporating label information to improve model specification and subspace learning, and constructing a heterogeneous learnware dock system that uses pseudo labels to encode model capabilities. Extensive experiments show that the recommended heterogeneous learnwares outperform user self-training with limited labeled data and improve task performance as more labeled data becomes available.

**Strengths:**

1. Addresses a practical and important problem of handling heterogeneous models with diverse feature spaces in the learnware paradigm, which is often encountered in real-world scenarios.
2. Innovative approach of leveraging label information, including model predictions and user's minor labeled data, to enhance the model specification and subspace learning, going beyond previous methods that relied on raw data or auxiliary co-occurrence data.
3. Thorough experimental evaluation on real-world tasks, demonstrating the significant performance improvements over user self-training with limited labeled data.

**Weaknesses:**

1. The potential challenges or limitations of the proposed model are suggested to be given.

2. The paper does not provide a thorough analysis of the computational complexity and scalability of the proposed framework, which could be important for large-scale, real-world deployments.

3. The experiments are conducted on a single dataset, and a more diverse evaluation of different types of tasks and datasets would strengthen the generalizability of the findings.

**Questions:**

None

---

> ### Author Rebuttal · Authors · 2024-08-06
>
> **Q1: The potential challenges or limitations of the proposed model are suggested to be given.**
>
> Thanks for your advice. We discuss the limitations of the proposed method in the checklist (see Q2: limitations).
>
> In this paper, we consider scenarios where the feature spaces of models in the learnware dock system are heterogeneous. However, real-world applications may also have heterogeneous label spaces. We will add more discussions on this in the revised version and please see the author rebuttal for a brief discussion.
>
> ---
>
> **Q2: The paper does not provide a thorough analysis of the computational complexity and scalability of the proposed framework, which could be important for large-scale, real-world deployments.**
>
> The process of the learnware paradigm is outlined as follows:
>
> * **Building the Learnware Market:** Developers generate specifications, package them with models into learnware, and submit them to the learnware dock system. This system organizes the learnware and assigns system-level specifications to the models.
> * **Utilizing the Learnware Market:** Users generate requirements and submit them to the system, which then recommends appropriate learnware. Users can then reuse the learnware.
>
> Below, we analyze the complexity of each process. The relevant symbols are defined as follows: the size of the raw dataset is $ n $, the number of categories is $ c $ (1 for regression), the dimension is $ d' $, and the size of the specification is $ m $. There are a total of $ N $ learnwares in the learnware dock system, and the complete feature space contains $ K $ sub-blocks with a total dimension of $ d $.
>
> **1. Specification/Requirement Generation**
>
> * **Initialization:** k-means clustering, with a complexity of $ O(nmcd') $.
> * **Iterative Optimization:** Achieved through alternating optimization, with the bottleneck being the inverse matrix computation of the kernel matrix, which has a complexity of $ O(cn^3) $.
>
> The total complexity is $ O(c(nmd' + T\_s n^3)) $. Typically, the number of iterations $ T\_s $ is small, around 10, and the size of the specification $ m $ is much smaller than the dataset size $ n $. When the sample size $ n $ is large, the complexity of specification generation is $ O(cn^3) $, scaling cubically with the size of the raw dataset $ n $. However, **with GPU acceleration, the time required for specification generation is quite small. For the dataset openml__volkert__168331 with shape (58310, 180), the generation time of specification is 2.826s (A100 80G*1).**
>
> **2. Subspace Learning**
>
> * **Single Epoch, Single Specification:** The complexities for calculating the contrastive loss, reconstruction loss, and supervised loss are $ O(m^2 K^2 d) $, $ O(mK d^2) $, and $ O(mcd + md^2) $ respectively, totaling $ O(m^2 K^2 d + mK d^2) $. The complexity of updating the model using the loss functions (both are two-layer fully connected networks) is $ O(m d^2) $. Thus, the complexity for subspace learning with a single specification is $ O(m^2 K^2 d + mK d^2) $.
> * **All Epochs, All Specifications:** Considering all specifications and the number of iterations, the complexity for subspace learning is $ O(N T\_{sub} (m^2 K^2 d + mK d^2)) $.
>
> **This complexity is linearly related to the number of learnwares $ N $ in the market and quadratically related to the size of the specifications $ m $，the dimension $ d $ of the feature space and the number of feature blocks $K$.**
>
> **3. Learnware Recommendation**
>
> The time complexity is $ O(Ncm^2) $, **linearly related to the number of learnwares $ N $ in the market**.
>
> **4. Model Reuse**
>
> When reusing a model, it only requires mapping the raw data through a two-layer fully connected network to the feature space corresponding to the heterogeneous learnware and making predictions using the learnware, which has low time complexity.
>
> **5. Summary**
>
> **The most time-consuming procedure of the learnware paradigm is the subspace learning for heterogeneous learnware recommendation. Even with a large number of learnwares $ N $, the total training data $ Nm $ for subspace learning remains manageable due to the small size of each specification (typically $ m = 50 $). Furthermore, our methods train an encoder and decoder for each feature block instead of each model. Although the number of models can be quite large, the number of feature blocks is relatively small.** An extreme case involves $ K $ blocks, resulting in $ 2^K $ feature spaces and $ 2^K $ models for each feature space, our method trains $K$ encoders and decoders, not $2^{K}$.
>
> Besides the theoretical analysis, we will also provide the actual running time in the attached PDF to demonstrate the efficiency of the proposed framework. Our method have much less time for preparation (construct the learnware market) compared to the pre-training methods, and the time for utilization (recommend and reuse learnware) is also efficient. Please see author rebuttal for details.
>
> ---
>
> **Q3: The experiments are conducted on a single dataset, and a more diverse evaluation of different types of tasks and datasets would strengthen the generalizability of the findings.**
>
> We have conducted experiments on real-world projects, please see author rebuttal for details. Thanks for your suggestion.

---

> > ### Comment · Reviewer_8wdE · 2024-08-11
> >
> > Thanks for the authors' rebuttal, I do not have any questions.

---

> > > ### Author Response · Authors · 2024-08-12
> > >
> > > Thank you again for reviewing our manuscript, your suggestions are valuable to us. We hope that our response adequately addresses your concerns. Please kindly let us know if you need any further clarification and we are prepared to provide any additional information that might be helpful. We would also greatly appreciate it if you could consider increasing the score.

---

### Official Review · Reviewer_NnRb · 2024-07-12

**Soundness:** 3
**Presentation:** 3
**Contribution:** 3
**Rating:** 6
**Confidence:** 4

**Summary:**

The paper focuses on the learnware paradigm and finds that label information plays an important role in it, which is both practical and interesting. It proposes a new specification that enhances subspace learning and improves learnware management. Extensive experiments demonstrate the superiority of the proposed methods.

**Strengths:**

1.The paper is well-written and easy to follow.

2.The paper addresses a significant gap in the learnware paradigm by proposing a method for handling heterogeneous feature spaces, a common real-world scenario.

3.The experiments are thorough, covering both classification and regression tasks, and comparing the proposed method against a wide range of baseline methods.

4.The code is provided in the appendix, enhancing the reproducibility of the results.

**Weaknesses:**

1.The method seems complex. Thus, a running time comparison is necessary to show its efficiency.

2.I wonder if the method can be used for larger data to show its scalability.

**Questions:**

Please see weaknesses.

**Limitations:**

The authors could clarify the limitations and future work for better understanding.

---

> ### Author Rebuttal · Authors · 2024-08-06
>
> **Q1: The method seems complex. Thus, a running time comparison is necessary to show its efficiency.**
>
> Thank you for your suggestion. In the author rebuttal, we provided a brief discussion; here is a more detailed explanation.
>
> **1. Evaluation Setup**
>
> The evaluation setup includes the preparation and utilization phases. Since the preparation phase is done once by non-user entities, we focus on the utilization phase time required for user tasks. The evaluation setup for each method is as follows:
>
> - **lightgbm**: Utilization phase only, training from scratch on user-labeled data.
> - **TabPFN, Transtab, Xtab**: Both phases. Preparation involves obtaining a pre-trained model; utilization involves fine-tuning on user data.
> - **Align_unlabeled, Align_labeled, Hetero, Our_unify, and Our_cls**: Both phases are involved. Preparation includes constructing and organizing a model hub, while utilization involves generating task requirements, recommending models, and reusing models on user data.
>   - Our_unify requires a self-trained model for pseudo-labeling during specification generation and is influenced by the model type and training parameters. Therefore, $\textbf{Our}_{\textbf{unify}}$ records the time assuming a pre-existing self-trained model, whereas $\textbf{Our}\_{\textbf{unify}}^{\text{lightgbm}}$ includes the entire process, including model training time using the time-consuming lightgbm method. Users can opt for a simpler model to reduce time costs.
>
> **2. Summary of Evaluation Results**
> - Classification Tasks:
>   - $\textbf{Our}\_{\textbf{unify}}$ ranks 1st, $\textbf{Our}\_{\textbf{cls}}$ ranks 3rd
>   - $\textbf{Our}\_{\textbf{unify}}^{\text{lightgbm}}$ ranks 7th (lightgbm ranks 5th).
> - Regression Tasks:
>   - $\textbf{Our}\_{\textbf{unify}}$ ranks 2nd
>   - $\textbf{Our}\_{\textbf{unify}}^{\text{lightgbm}}$ ranks 4th (lightgbm ranks 3rd).
>
> **3. Analysis of Utilization Phase**
>
> Compared to other methods with $\textbf{Our}_{\textbf{unify}}$:
> - **lightgbm**: Requires training a model from scratch, whereas our methods reuse relevant heterogeneous models, making them faster.
> - **TabPFN, Transtab, Xtab**: Our methods do not need the entire labeled user data for fine-tuning, making them quicker.
> - **Align_unlabeled, Align_labeled**: Our methods do not traverse the model hub but directly find the most suitable model, making them faster.
> - **Hetero**: Similar process, but Hetero might recommend multiple models, whereas our methods recommend only one, making them faster.
>
> Among our methods, $\textbf{Our}\_{\textbf{unify}}$ is faster than $\textbf{Our}\_{\textbf{cls}}$. $\textbf{Our}\_{\textbf{unify}}$ compresses X first, then uses the model to predict on the reduced set, generating the specification quickly if user model is already prepared. $\textbf{Our}\_{\textbf{cls}}$ predicts first, then compresses X and Y distributions simultaneously, making it slower.
>
> **By efficiently generating user task requirements and quickly reusing heterogeneous models, our method reduces the utilization phase time compared to other methods.**
>
> **4. Analysis of Preparation Phase**
> - **TabPFN**: Pre-training from simulated datasets generated by structure causal model, learning the posterior distribution $P(y\_{test}|x\_{test},D\_{train})$ and applying it to small-scale real tasks. Pre-training time: 20 hours (8*RTX2080Ti).
> - **Xtab**: Learns a shared backbone network from many tasks; pre-training time unknown.
> - **Transtab**: Uses feature descriptions and values from related tasks to generate a unified subspace, training a shared Transformer backbone. Average pre-training time: 672 seconds (classification) and 1425 seconds (regression).
> - **Align_unlabeled, Align_labeled**: No model library organization time.
> - **Hetero**: Organization time: 2.44 seconds (classification) and 2.55 seconds (regression).
> - **Our_unify**: Organization time: 41.39 seconds (classification) and 40.78 seconds (regression).
> - **Our_cls**: Organization time: 45.69 seconds (classification).
>
> Times are based on a single A100 GPU.
>
> **Compared to methods using pre-trained models, our methods significantly reduce preparation time by not requiring training on complete datasets but using specifications, which are smaller. We also do not train a unified backbone network, only performing simple subspace learning.**
>
> ---
>
> **Q2: I wonder if the method can be used for larger data to show its scalability.**
>
>
> Thank you for your question. Indeed, our experiments test datasets of various sizes and show that even with more labeled data, our paradigm enhances user performance. We also discuss the scalability of the learnware dock system with numerous heterogeneous learnware in the author rebuttal.
>
> **1. Our experiments test datasets of various sizes**
>
> We tested datasets with varying sizes. For classification tasks, sample sizes range from 1,000 to 58,310, feature dimensions from 7 to 7,200, and classes from 2 to 10. For regression tasks, sample sizes range from 418 to 108,000, with feature dimensions from 8 to 128.
>
> **2. Our method boosts performance even with more labeled data**
>
> Tables 1 and 2 in our paper show that with only 100 labeled samples, using the recommended heterogeneous model significantly outperforms user self-training. Figures 4 and 5 illustrate that combining the heterogeneous model with user self-trained models improves performance across different labeled data amounts. **On average, with 500 labeled samples, reusing the recommended model still outperforms self-training. With 2,000 labeled samples, the heterogeneous model continues to enhance performance. Even with 5,000 labeled samples, learnware improves 21% of classification cases and 50% of regression cases.** In some cases, like kin8nm, even using the entire training dataset, the heterogeneous model improves user performance.
>
> **3. The cost of organizing a learnware dock system with numerous heterogeneous learnware is relatively small**
>
> Please refer to the author rebuttal for details.

---

> > ### Comment · Reviewer_NnRb · 2024-08-08
> > **Response**
> >
> > Thank you for the rebuttal. As my original scoring is optimistic, I retain my scoring to this paper with higher confidence.

---

> > > ### Author Response · Authors · 2024-08-08
> > >
> > > Dear Reviewer,
> > >
> > > Thank you for increasing your confidence level to 4 (confident but not absolutely certain). We greatly appreciate your thorough review of our manuscript and your valuable suggestions. If you have any further questions, please feel free to ask. We are ready to provide any additional information that might be helpful.

---

### Author Rebuttal · Authors · 2024-08-06

Dear Reviewers,

**Please see the attached one-page PDF with a summary of additional experimental results regarding time analysis and performance on real-world projects.**

We would like to thank all reviewers for their constructive feedback, which has greatly improved our paper. We are encouraged by the reviewers' positive remarks, including:

* The **practical and important problem** of handling learnwares from heterogeneous feature spaces (Reviewer NnRb, 8wdE) and the recognition of learnware as a useful learning paradigm (Reviewer PBMH).
* The **novel observation** that matching with only the marginal distribution is insufficient (Reviewer HLyF).
* The well-written and **easy-to-follow** paper (Reviewer NnRb, HLyF), with thorough (Reviewer NnRb) and good (Reviewer HLyF) experiments .

---

We have diligently worked on addressing your suggestions and have summarized the additional experiments and analysis below.

**1. Running Time and Scalability**

**Running Time.**  Our experiments demonstrate the efficiency of our method through comparative analysis (see Tables 1 and 2 in the attached PDF). The evaluation is divided into the preparation phase (pretraining a model/building the model hub) and the utilization phase (fine-tuning the pre-trained model/recommending and reusing models), with a primary focus on the latter as it involves user tasks. **When a user has a self-trained model for pseudo-labeling a reduced set during specification generation, our method** $\textbf{Our}\_{\textbf{unify}}$ **ranked 1st in classification tasks and 2nd in regression tasks.**  Even when including our method with a self-training procedure using the time-consuming LightGBM, $\textbf{Our}\_{\textbf{unify}}^{\text{lightgbm}}$, it is still much quicker than most pretrained methods (Transtab, Xtab) and the model hub method that requires fine-tuning the model $\textbf{Align}\_{\textbf{label}}$. **The efficiency of our method is heavily based on specifications, which are much smaller than the original dataset.**  Specifications quickly sketch the user task and help match the relevant heterogeneous learnware, while the mapping functions learned during the learnware dock system construction help quickly reuse heterogeneous learnware. **All procedures avoid using the whole dataset for fine-tuning.** In the preparation phase, our method is much quicker than pre-trained methods (detailed results in the Response to Reviewer NnRb), because our method only reuses specifications for subspace learning with simple mapping functions, rather than training a shared, complicated Transformer-based backbone with raw data.

**Scalability.**  In this part, we talk about the case when the learnware dock system have too many heterogeneous learnware. The most time-consuming aspect of the learnware paradigm is the subspace learning required for heterogeneous learnwares organization. **Even with a large number of learnwares $N$ ​, the total training data $Nm$ for subspace learning remains manageable due to the small size of each specification (typically $m=50$​ ).** Additionally, our methods **train an encoder and decoder for each feature block rather than for each model**.  Despite the potentially large number of models, the number of feature blocks is relatively small. For $K$ blocks resulting in $2^K$ feature spaces and models, our method trains only $K$ encoders and decoders, not $2^K$. This also helps address user tasks with combinations of feature blocks not present in the learnware dock system.

**2. Performance on Real-World Projects**

We tested our method on two real-world projects: a hot sales-forecasting competition (Predict Sales Forecasting, short for PFS) on Kaggle, a regression task with raw data containing 6 tables, and a widely used large clinical database of critical care units called MIMIC-III [Johnson etal., 2016], which contains 26 tables.

For PFS, we selected a popular feature engineering in the competition, and for MIMIC-III, we used the MIMIC-III benchmark [Harutyunyan et al., 2019]'s preprocessing and selected the in-hospital mortality task, a binary classification task. For processed data, we split the feature space according to its semantics for PFS and randomly split for MIMIC-III due to its large feature space dimension (714). For PFS, we further split the data by location, as sales forecasting is evaluated locally. The data for each location is at the hundred thousand level.

**Our methods ranked first, demonstrating superior performance over other contenders in both real-world medical classification and business regression tasks.**

---

Finally, we discuss the extension of our method. Although our method is designed for heterogeneous feature spaces, it can also be naturally extended to handle both heterogeneous feature and label spaces. We give a brief discussion as follows:

**3. Extension for heterogeneous feature & label spaces**

We assume that the overall feature space and label space for a task are represented as $\mathcal{X}=\mathcal{X}\_i\times\cdots\times\mathcal{X}_Q$ and $\mathcal{Y}$, respectively. When the learnwares have heterogeneous feature spaces $\mathcal{X}\_i=\times\_{k\in C\_i}\mathcal{X}\_k$ and label spaces $\mathcal{Y}\_i\subseteq\mathcal{Y}$, our method can be naturally extended to address this problem. We present the following extension as an example. The subspace can be learned in the same way, **with the supervised term playing an important role in label space alignment within the subspace**. For learnware recommendation, the learnware dock system can recommend the most suitable model **for each class** of the user's task using the MMD distance. For the recommended learnwares, reuse can be achieved through **dynamic classifier selection**, as demonstrated in previous work (TKDE 2023).

---

For other questions, please refer to our reviewer-specific feedback for further details. We hope our responses are satisfactory and can address your problem.

---

### Decision · Program_Chairs · 2024-09-25

**Decision:**

Accept (poster)

**Comment:**

The proposed methods have been shown to improve SOTA on tasks with heterogeneous feature spaces, availability of some labelled data, and requirement for non-exposure of raw data of the model developers. While this is a relatively specific setting, the manuscript together with the rebuttal have sufficiently convincingly explained why this is a relevant task for the community.

The submitted version of the paper did not consider a realistic heterogeneous feature space setup, but instead was randomly dividing the data sets. However, during the rebuttal phase, the authors added experiments about two real-world scenarios, and in one of those the split was designed to be non-random and more realistic.

All 4 reviewers have stated that the presentation was good. All reviewers also see the paper as sound, except for some concerns of one reviewer, but these were alleviated during the rebuttal.

Further analysis of method complexity was requested by one of the reviewers, and this was provided during the rebuttal.

Overall, the paper can be recommended for acceptance.